# A single-cell resolved genotype-phenotype map using genome-wide genetic and environmental perturbations

Mariona Nadal-Ribelles [1,2] ✉, Carme Solé [1,2], Anna Díez-Villanueva[2], Camille Stephan-Otto Attolini [2], Yaima Matas [1,2], Lars Steinmetz [3,4], Eulàlia de Nadal [1,2] ✉ & Francesc Posas [1,2] ✉

Heterogeneity is inherent to living organisms and it determines cell fate and phenotypic variability. Despite its ubiquity, the underlying molecular mechanisms and the genetic basis linking genotype to-phenotype hetero-geneity remain a central challenge. Here we construct a yeast knockout library with a clone and genotype RNA barcoding structure suitable for genome-scale analyses to generate a high-resolution single-cell yeast transcriptome atlas of 3500 mutants under control and stress conditions. We find that transcriptional heterogeneity reflects the coordinated expression of specific gene programs, generating a continuous of cell states that can be responsive to external insults. Cell state plasticity can be genetically modulated with mutants that act as state attractors and disruption of state homeostasis results in decreased adaptive fitness. Leveraging on intra-genetic variability, we establish that regulators of transcriptional heterogeneity are functionally diverse and influenced by the environment. Our multimodal perturbation-based single-cell Genotype-to-Transcriptome Atlas in yeast provides insights into organism-level responses.

Variability within a population determines cell fate and phenotypic diversity over time (e.g. differentiation, aging)[1] and facilitates population adaptation to different niches and conditions. Remarkably, even genetically identical cells display variations in growth rate, division, stress resistance, and other quantifiable phenotypes, which lead to global health challenges, such as tumor aggressiveness, chemotherapy resistance, aging, and antibiotic resistance in microorganisms[2,3]. Resolving the transcriptome profile of individual cells has been essential in enabling Cell Atlas initiatives seeking mainly to determine the expression programs that identify new cell types and define subtypes and cell states[4]. Identifying the molecular basis that defines cell states requires exhaustive cell profiling under both homeostasis and perturbation, an endeavor that will provide the principles to target or engineer specific cell states and, therefore, determine cell population fate. The concept of cell state has evolved beyond the cell cycle to reflect more dynamic complex cellular behaviors, which are increasingly recognized as being relevant in the transition from health to disease (quiescence, senescence, resistant phenotypes...)[5–7]. While our ability to profile the transcriptomes of individual cells has increased at unprecedented speed, the molecular basis underlying transcriptional heterogeneity, which defines and regulates phenotypic diversity and plasticity, remains a central challenge.

Genetic screens and extracellular condition profiling have been used to define the relationship between transcriptional heterogeneity and phenotype, mostly using CRISPR knockdown screens in cancer cell lines[8–11]. Yeast is an ideal model for assessing transcriptional heterogeneity at the organismal level. In this regard, *Saccharomyces cerevisiae* has pioneered functional screens due to the vast number of

[1]Department of Medicine and Life Sciences, Universitat Pompeu Fabra, Barcelona, Spain. [2]Institute for Research in Biomedicine (IRB Barcelona), the Barcelona Institute of Science and Technology, Barcelona, Spain. [3]Department of Genetics, Stanford University, School of Medicine, California, USA. [4]European Molecular Biology Laboratory, Heidelberg, Germany. ✉e-mail: mariona.nadal@irbbarcelona.org; eulalia.nadal@upf.edu; francesc.posas@irbbarcelona.org

molecular tools and genetic resources available for this organism. The amenable yeast genome allows the generation of isogenic genetic perturbations, thereby avoiding the variability generated by CRISPR knockouts or knockdown efficiency using CRISPR inactivation (CRISPRi)[12,13]. One of the most revolutionary tools is the yeast knockout collection (YKOC), in which non-essential genes are deleted from the genome. The YKOC served to dissect genotype-phenotype relationships in bulk assays and has set the groundwork for our understanding of gene network analyzes[14,15]. While the phenome of the YKOC has been extensively studied (Yeast Phenome[16]), at the transcriptomic level the large-scale reference dataset contains only a bulk analysis from 25% of the non-essential deletions. At the single cell level, and unlike higher eukaryotic systems which often rely on CRISPR for genetic perturbations, in yeast an array of 12 transcription factor mutants has been profiled by manually creating and validating expressible barcoded deletions across multiple conditions[17]. However, despite the remarkable ease of genomic engineering in yeast, surveying large-scale genetic perturbations at the single-cell level has been hindered by the absence of technologies that can systematically integrate with single-cell RNA sequencing (scRNA-seq)[18].

To bypass these limitations, we developed a yeast single cell resolved genotype-to-transcriptome atlas. To this end, we created a RNA-barcoded genotype and clone deletion collection by reconfiguring the classical YKOC, to generate a stable clone and genotype barcoded collection suitable for both targeted and large-scale single cell perturbation studies. The design enabled us to generate two transcriptome-wide single-cell atlases with genome-scale perturbations under control and stress conditions. The dataset provides high resolution transcriptional profiles of over 3500 mutants covering 75% of the non-essential genes in the organism. The comprehensive nature of our study provides a reference map of the yeast cell state space and genetic configuration, but also elucidates the influence of genetic perturbations on transcriptional heterogeneity, thereby revealing the underlying genetic basis of transcriptional heterogeneity.

## Results

### Redesign of the YKO collection with RNA-traceable deletions to enable genome-scale genetic and environmental perturbation screens

To characterize the relationship between genotype and transcriptome at single cell resolution, we generated a genome-scale library of RNA-traceable deletion mutants by reengineering the yeast knockout collection (YKOC). The YKOC includes deletions of most non-essential genes in the *S. cerevisiae* genome. The structure of the gene deletion cassette to knockout (KO) each gene by homologous recombination consists of a constitutive promoter (pTEF1) that drives the expression of *(i)* the *KANMx4* resistance gene and *(ii)* two unique genotype barcodes (20 bp) per deletion; both flanked by common sequences upstream (Uptag) and downstream (Downtag) of the promoter and terminator respectively[14] (Fig. 1a; Methods). This approach renders the genotype identity invisible to transcriptomic readouts and incompatible with single-cell genetic perturbation screens (Perturb-seq). Inspired by the original YKOC gene deletion cassette, we redesigned its structure to generate an RNA-traceable clone and genotype for each mutant (Fig. 1a). Briefly, we generated a PCR cassette to replace the *KAN* resistance marker with *URA3*. This cassette serves two purposes. First, it shortens the heterologous terminator linking the original Downtag barcode to the 3'UTR of *URA3* to a minimum (43 nt) and, second, it adds a clone barcode (5 random nucleotides) downstream of the *URA3* STOP codon (see Methods). By doing so, the 3'UTR of *URA3* provides a mean to label and transcriptionally trace genotypes and clones (Fig. 1a, Supplementary Fig. 1a). In contrast to pooled CRISPR perturbations, to generate the YKOC each mutant was grown and transformed individually, which prevents competition between mutants, and positive clones were selected by successive rounds in

selective media (see Methods). The final collection consisted of 4162 mutant strains (82% of the original YKOC) carrying RNA-traceable barcodes suitable for genome-scale genetic screens combined with single-cell transcriptomics (Perturb-seq). This unique library represents a resource from which the same perturbations can be tested elsewhere, but it is also flexible to allow the user to assess specific mutants of choice. Of note, having fully independent clones helps test how a given mutation performs across individual yeast cells allowing for the normalization across epigenetic yeast individual variability, but also permits to interrogate how perturbations affect such individual epigenetic variability.

We designed two genome-scale Perturb-seq in which the whole collection of mutants was grown independently (96-well plates) and pooled before being subjected to stress (osmostress; 0.4 M NaCl, 15 min) or control conditions (Fig. 1b). Due to the rapid and transient response of cells to osmostress, we collected and methanol-fixed cells at the peak of the transcriptional response to preserve the transcriptome (see Methods)[19,20]. To generate the scRNA-seq libraries, we used a microwell-based platform for single-cell isolation and oligo dT priming to capture polyadenylated RNA for cDNA synthesis (Yeast GENEXSCOPE HD, Singleron Biotechnologies). We profiled a total of 1.061.865 cells, from which we removed low-quality ones (± 2 standard deviation of mean genes, >10% mitochondrial reads, see Methods). In parallel, for each library, we performed a one-step PCR amplification of the clone and genotype barcode from the *URA3* 3'UTR (Supplementary Fig. 1b; see Methods). We combined the transcriptome and targeted amplification to assign genotype identity for more than 71% of cells, removing cells with both conflictive and unassigned genotypes from the analysis (Supplementary Fig. 1c). The resulting dataset contained more than 3500 mutant genotypes (Fig. 1c) equally represented across conditions (Fig. 1d) with an average of 93 and 108 cells/genotype (median of 77 and 87 respectively) in control and stress conditions, respectively, and had a median coverage of 550 genes/cell and 1200 molecules/cell. Mapping the 3'UTR reads downstream of the genotype barcode allowed us to validate the genomic location of the deleted genes. Overall, 90% of genotype barcodes aligned closely with the endogenous terminator of the deleted gene, thereby confirming genomic loci and thus the robustness of the data (Supplementary Data 1). Correspondingly, we observed a reduced expression of the knockout gene in the assigned genotype (Fig. 1e). We incorporated clone barcodes into the YKOC to extend its functionality for bulk and single-cell applications. Of note, clones of genotypes with high clonal coverage ( > 10 cells per clone), displayed a similar transcriptome with 77% of the clones displaying <3 differentially expressed genes, suggesting that clonal differences are small (Supplementary Fig. 1d). Thus, the library now permits clonal analyzes, allowing comparisons and deeper analyzes of single cell heterogeneity.

To generate the transcriptome atlas, we estimated the global effect of each mutant by comparing its transcriptome against the wild type in the corresponding condition. In control conditions, our data indicates that most mutants exhibit a pattern of differential expression favoring upregulated genes (Fig. 1f, Supplementary Fig. 1e,f). Approximately 50% of the mutants have more than 10 differentially expressed genes, with 10% having a very strong transcriptional phenotype (Fig. 1g). These data align with the largest transcriptomic profiling in bulk ( > 1400 mutants), where most deletions also biased towards upregulation of gene expression[21]. However, this trend fades under stress conditions, resulting in a balanced outcome with mutants displaying both upregulation and downregulation of gene expression (Fig. 1h,i, Supplementary Fig. 1g,h). To validate the robustness of the data, we compared our scRNA-seq dataset to the largest bulk transcriptome profiling of deletions in yeast using microarrays[21]. Despite substantial methodological differences, our dataset showed a consistent correlation between the number of differentially expressed genes per genotype in both studies (Supplementary Fig. 1i). Similarly,

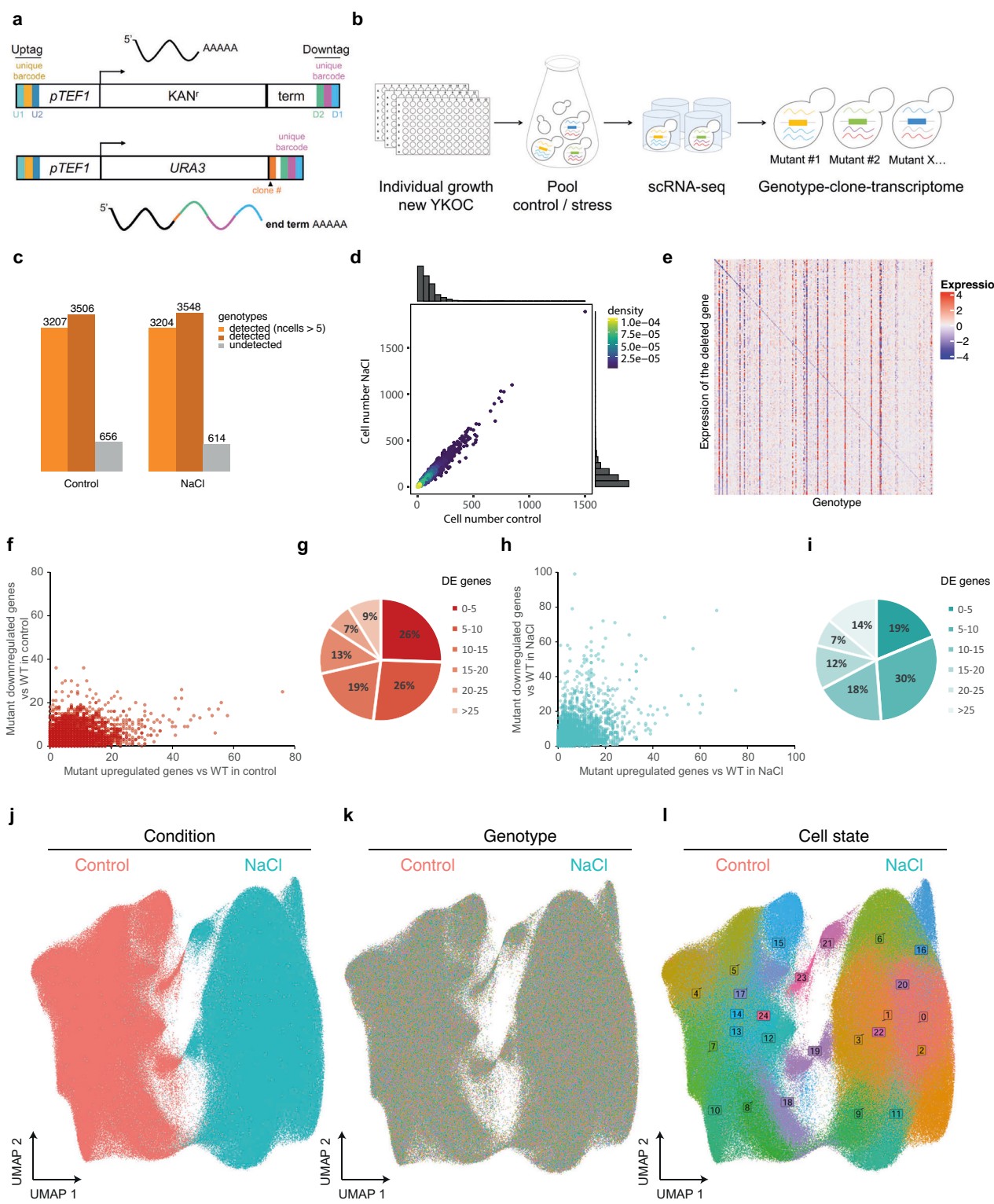

mutants known to have an impaired transcriptional response ranked among the lowest expressing mutant in agreement with previous reports[22,23] (Supplementary Fig. 1j). Overall, these results indicate that our scRNA-seq dataset recapitulates bulk transcription patterns. Additionally, these results provide an overview of the functional impact of genetic mutations and highlight the relevance of the external conditions.

To identify sources of intrinsic heterogeneity and avoid normalization biases, we regressed the cell cycle and removed ribosomal genes from the expression matrix as these are known to introduce technical noise and are commonly accounted for in scRNA-seq analysis[2,24]. We used highly variable genes as an input to represent the transcriptome of single-cells using Uniform Manifold Approximation and Projection (UMAP) embedding. This approach revealed stress condition as the predominant clustering factor over genotype identity, batch, and detectable before and after cell cycle regression (Fig. 1j,k, Supplementary Fig. 1k–m). To cluster cells based on their degree of transcriptional similarity, we applied the Louvain algorithm[25] to extract the transcriptional signature associated with each cell state. Interestingly, we observed that cell states within a population are robust to

**Fig. 1 | Redesign of the YKO collection with RNA-traceable deletions to enable genome-scale genetic and environmental perturbation screens. a** Schematic representation of the RNA-traceable YKOC. Structure of the original deletion (upper panel) and structure of the RNA-barcoded clone and genotype their position in the 3'UTR. **b** Experimental layout. Cells were grown individually before being pooled together and subjected or not to stress (0.4 M NaCl for 15 min). Fixed cells were used for microwell based scRNA-seq. The transcriptome of each cell was linked to the transcriptome through the *URA3* transcript (see methods). **c** Genotype coverage per each condition is shown based for all detected genotypes (dark orange), genotypes with >5 cells (light orange) and unassigned cells based on the 4162 genotypes of the RNA-barcoded collection. **d** Correlation of cell number per genotype across conditions. Points are colored by density, warmer colors indicate higher density. Marginal histograms show the distribution of cell numbers for each condition. The mean and median cell number per condition are shown in the corresponding axis. **e** Heatmap showing the expression levels of knocked-out genes in relation to assigned genotypes for the top 200 highest expressing genes.

Each row (y-axis) represents a knocked-out gene, while each column (x-axis) represents an assigned genotype based on barcode expression. Color intensity indicates gene expression level, warmer colors indicate representing higher expression. The diagonal blue pattern indicates reduced expression of each knocked-out gene in its corresponding genotype. **f** Scatter plot represents the comparison of upregulated (x-axis) and downregulated (y-axis) genes for each genotype (points) in control conditions compared to the wild type strain in control. **g** Fraction of genotypes in which genes are differentially expressed in control. **h** Scatter plot represents the comparison of upregulated (x-axis) and down-regulated (y-axis) genes for each genotype (points) in control conditions compared to the wild type strain in stress (NaCl 0.4 M 15 min). **i** Fraction of genotypes in which genes are differentially expressed in stress. **j–l** UMAP of the entire dataset across conditions (j), genotype identify (k) and cell states defined by Seurat (L). UMAPS represent the complete dataset of cells passing the quality check and assigned to a genotype (n = 710952 cells). Source data are provided as a Source Data file.

---

loss-of-function mutations (Fig. 1l). These results are concordant with a condition-dependent clustering recently reported in a targeted transcription factor scRNA-seq profiling grown is several conditions in yeast[17]. Thus, we developed a robust framework that enabled us to build a systematic atlas of the single-cell transcriptome response to genetic and environmental perturbations of more than 3500 mutants under control and stress conditions.

## Cells in a population organize into heterogeneous gene expression states

Given the large-scale dimension of the dataset and the important contribution of cell states, we clustered each condition independently. In the UMAP space, cells distributed into 20 states in control (0–19 C) and 18 states in stress conditions (0–17 N) (Fig. 2a,b). In both conditions, cells mainly organized into a central cluster defining a continuum of cell states. We identified the genes that define transcriptional states by extracting state markers and assign function using upregulated genes (log2FC > 0.25 and *p-value* < 0.05) (Fig. 2c, Supplementary Data 2,3). Interestingly, these markers were not markedly distinct, and several genes were shared across states, thereby pointing to a continuum of states within a cell population. In addition, we found that the wild type and, remarkably, 90% of the mutant strains, in both conditions, shared the same distribution across states (Supplementary Fig. 2a–d). Most cell cycle phases were distributed across cell states; however, despite cell cycle regression, we identified one cluster (10 C and 11 N) enriched in S phase cells, characterized by high expression of histone genes (Supplementary Fig. 2e,f and Supplementary Data 4). Therefore, our data suggests that several cell states represent stable and defined transcriptional states rather than genotype-specific states and thus, extremely robust against genetic perturbations.

In control conditions, cell states involved a variety of functions encompassing the following: cell morphogenesis (e.g., cell wall biogenesis, cell conjugation, pheromone signaling and agglutination genes); metabolic processes (e.g., carbon metabolism, phosphate metabolism, iron metabolism and mitochondrial genes); and protein homeostasis, among others (Supplementary Data 2,3). Three distinct clusters fully separated from the core UMAP, and these displayed the upregulation of: the environmental stress response (ESR) (14 C), aging-related signatures (15 C), and mating genes (16 C) (Fig. 2a). Of note, clusters 16 C, 2 C and 12 C (31% of the population) displayed gene signatures of mating pheromone signaling pointing to this pathway as an important contributor to cell state diversity (Supplementary Fig. 2g). Indeed, the heterogeneous expression of pheromone signaling genes, including the detected cell state markers (e.g. *FIG*, *AGA* genes), has been found in a similar percentage of the population under normal conditions using nascent transcription reporters, thereby reinforcing the robustness of our dataset[26].

Among the cell state markers, we recognized well-known transcription programs, such as those that define daughter cells, characterized by the expression of daughter-specific genes (*DSE1-4*, among others[27]. Of note, based on the expression of the *DSE* genes, we distinguished two subpopulations of daughter cells with a differential degree of development; one of them shows the described co-regulation of the *DSE1,2* pair with *CTS1* (the chitinase responsible for degrading the mother-daughter barrier) that identifies the most naive daughter cells (3 C) versus a second population in which cells expresses the *DSE3,4*-genes together with the mother early G1 marker *PIR1-HSP150* (1 C) (Supplementary Fig. 2g)[17], thus recapitulating known yeast developmental stages.

Remarkably, clusters 8 C and 19 C showed a strong upregulation of the iron regulon (*FIT2*, *FIT3*, *FET3*, *ARN1*, among others), mirroring the hallmark signature of aged cells defined by bulk RNA-seq[28] (Fig. 2d) and correspondingly, accumulated iron[28]. Of note, while 8 C highly expressed the iron regulon, 19 C had fewer cells but also expressed a larger number of genes involved in cell wall organization induced during stress conditions, perhaps representing a later aged stage and allowing us to trace different degrees of phenotypical development. Moreover, extending the aging signature defined in bulk to the top 10 genes and distinguishing between iron-related and unrelated genes, clusters 4 C and 15 C also stood out, revealing a bimodality within aging states in the cell population. This second module (4 C and 15 C) included the upregulation of genes known to be induced under hypoxia (*TIR*, *DAN* and *PAU* families) but not the upregulation of iron genes (Fig. 2d and Supplementary Fig. 2h). Thus, the scRNA resolution allowed us to resolve the bulk aging signature into two distinct transcriptional programs underlying two different aging subtypes via modular expression of iron (represented by *FIT3*) or hypoxic genes (*HUG1*) indicating that aging is a transcriptionally defined cell state that can occur via different pathways.

To confirm whether transcriptional signatures indeed reflect an aged population, we generated three destabilized fluorescent reporters (mCherry) driven by the expression of iron or hypoxia related promoters and terminators and a reporter from the pheromone signaling pathway as a control of an unrelated cluster (16 C) (*FIT3*, *HUG1* and *SAG1* respectively). We isolated the top 2% expressing population of each reporter by Fluorescence Activated Cell Sorting (FACS) (Supplementary Fig. 2i) and measured their replicative age (number of cell divisions) by counting bud scars stained using Calcofluor White stain and measured mitochondrial morphology using MitoTracker staining compared to a randomly sorted wild type expressing a constitutive mCherry reporter (p*TEF1*). Correspondingly, cells expressing either of the differential aging reporter genes (i.e. *FIT3* or *HUG1*) but not the *SAG1* reporter, showed an aged phenotype as seen by the number of scars compared to wild type (Fig. 2e), and an increase in fused mitochondria (Fig. 2f), both classical markers of aged cells[29,30]. Thus,

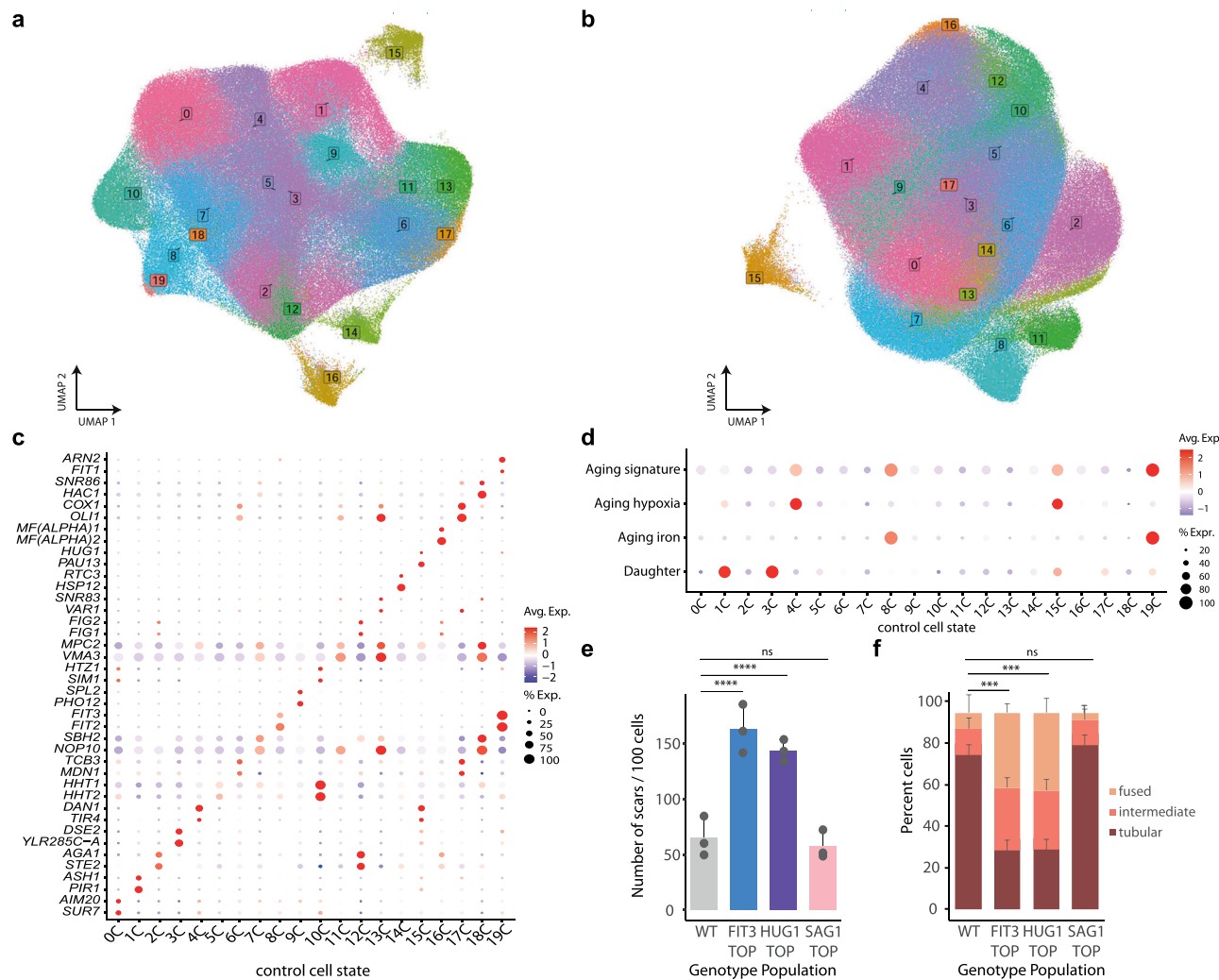

**Fig. 2 | Cells in a population arrange in heterogeneous gene expression states associated with biological function. a,b** UMAP of the control **a** and NaCl **b** dataset individually. Cells are colored by cell state indicated by boxed numbers colored according to the corresponding cluster. **c** Expression of two representative cell state marker genes in the control dataset. Dots size represent the percentage of expressing cells and are colored from high (red) to low (blue) expression. **d** Expression of the indicated aging signatures across control cell states. **e** Number of scars per 100 cells determined by Calcofluor White Stain for a wild type population or the top 2% of cells expressing the p*FIT3*- p*HUG1*-or *SAG1*-reporters. Data represents the mean and standard deviation of three independent replicates. *p*-values represent Benjamini–Hochberg corrected paired t.test against the wild type (*n* = 3) (*p*-value 0.0002 (*FIT3*), 0.0004 (*HUG1*), 0.5992 (*SAG1*)). **f** Distribution of mitochondrial morphologies for the indicated strains under control conditions by MitoTracker staining. Mean and standard deviation for each morphology category is shown (*n* = 3). (*p*-value 0.005 (*FIT3*), 0.004 (*HUG1*), 0.554 (*SAG1*)). Source data are provided as a Source Data file.

indicating that within a cell population, coordinated heterogeneous gene expression programs lead to interconnected cell states which can generate distinct cell fates. Taken together, our dataset captures and extends known cell states, providing the most comprehensive map of cell states defined by specific gene programs which reflect distinct cell fates within a population.

**Transcriptome mapping unveils core and responsive cell states**
To assess whether cell states are conserved in different conditions, we assessed the degree of transcriptional correlation across cell state markers across conditions. We classified "core states" as those whose gene expression patterns were reciprocally represented in both conditions (e.g., daughter, aged cells), while "responsive states" were defined as those differentially modulated depending on the condition (Fig. 3a,b).To assess the stability of cell states we took 2 approaches. First, we assessed the between control and stress clusters using upregulated cell state markers and found that a large proportion of

states in the control condition had at least one counterpart upon exposure to stress (Fig. 3c). We next performed a quantitative an unbiased approach using a label transfer approach to identify cell pairs that share the same transcriptional state across conditions (see Methods). We defined core states as those where at least 40% of cells in a control cell state paired uniquely with a stress cluster (Supplementary Fig. 3a, and Supplementary Data 5). Core states included developmental and morphogenic programs, cell cycle, the aging states, and states related to mitochondrial function. These observations thus indicate a high degree of conservation of the core clusters of gene expression across conditions.

Responsive states arose either because the expression of cell state markers are modulated upon stress thus losing their specificity, or because novel stress-regulated states appear during adaptation. For example, under stress, markers of the hypoxic aging and phosphate metabolism clusters (4–15C and 9 C, respectively) were no longer cluster-specific, suggesting altered cell wall composition and

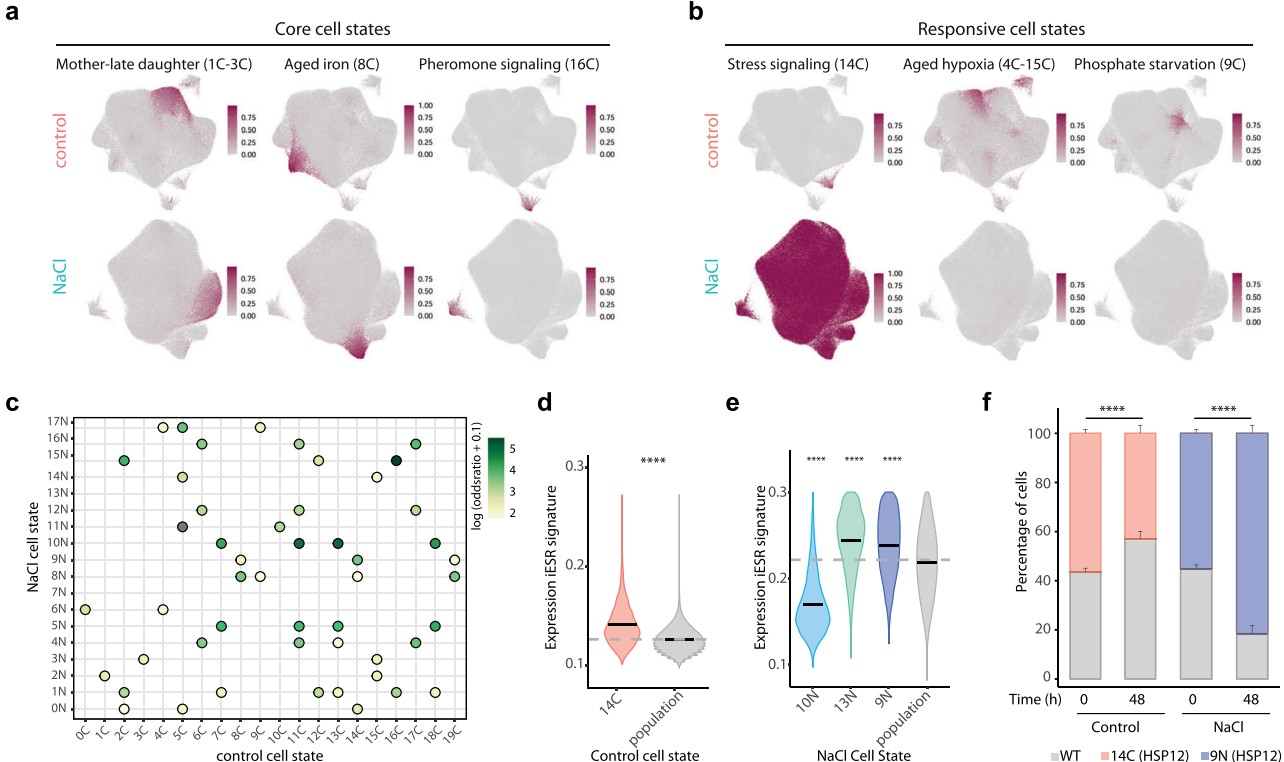

**Fig. 3 | Transcriptome mapping unveils core and responsive cell states that determine fitness. a,b** Examples of core and responsive cell states. UMAP shows the distribution of the cell state-specific signatures for the indicated clusters and condition. Cells are colored by degree of signature expression using two representative genes per cell state. Darker colors represent higher expression. **c** Pairwise cell state marker expression correlation across cell states. Per each cell state the expression of upregulated genes was correlated, dots colors represent the degree of similarity determined by Fisher test (darker color indicate higher correlation). **d,e** Expression of the iESR signature for the indicated cell states compared to the median of population (gray; and gray line). Stars indicate significance comparing against the median. (3 d, Wilcoxon test, <2e−16, 3e, Wilcoxon test <2e-16 for all groups. **f** Competition assay to determine cell state fitness. Cells with the top 2% expression of the *HSP12* or *SAG1* reporter were sorted and grown in combination with a wild type cell (labeled with GFP) in rich media (control) or in the presence of stress (1 M NaCl). Growth of the two populations was assessed at time 0 or after 48 h by flow cytometry. Data represents mean and standard deviation of three independent experiments and statistical significance is shown respect to the wild type strain (n = 3). Control *p-value* = 8,19388E-06, NaCl *p-value* = 8,64624E-05. Source data are provided as a Source Data file.

metabolism (Fig. 3b). On the other hand, some stress-dependent cell states showed a weak correlation with control states due to the expression of stress-related markers genes. These involved multiple functions, including carbon metabolism (clusters 0 N and 13 N), ADP metabolism (cluster 14 N), and protein folding (cluster 17 N), thereby suggesting that clusters that arise in the presence of stress are related to specialized protective functions (Fig. 3c, Supplementary Fig. 3b and Supplementary Data 2,3). For example, the stress cluster 9 N correlated with the basal-stress cluster in the control condition (14 C) and both displayed higher expression of the induced Environmental Stress Response (iESR) (e.g *HSP12*) identified by bulk RNA-seq[31,32] (Fig. 3d,e and Supplementary Fig. 3c).

We then assessed the phenotypic consequence of high expression of the iESR of cells residing in clusters 14 C and 9 N. To this end, we expressed a destabilized fluorescent reporter (p*HSP12*-iRFP-t*HSP12*) as mean to label these populations in control or upon stress or used the *SAG1* reporter (p*SAG1*-mCherry-t*SAG1*) as a control. We isolated the top 2% of cells expressing the reporter and tested their competitive fitness in comparison to wild type labeled with a constitutive mCherry reporter (p*TEF1*-mCherry-t*TEF1*) or GFP to compete against the top *HSP12* or *SAG1* expressor cells. The stress response program (14 C) exhibited a fitness trade-off: it reduced fitness under normal conditions but showed better fitness under stress (Fig. 3f). This suggests that the unnecessary expression stress-responsive genes impairs growth, yet provides a bet-hedging strategy under stress. In contrast, the top 2% of *SAG1*-expressing cells showed no change in competitive fitness

under either normal or stress conditions (Supplementary Fig. 3d). In addition, we noticed that cluster 13 N which was also hyperesponsive, was enriched in the expression of daughter-specific transcripts (Supplementary Fig. 3b). A global comparison of the response of mother and daughter cells revealed that the latter systematically showed a stronger induction of the iESR (Supplementary Fig. 3e), thereby suggesting that cell states determine the plasticity and resistance to different environmental conditions.

Therefore, the Yeast Transcriptome Atlas examined the transcriptional behavior of cell states under both stress and non-stress conditions for more than 3500 mutants. This allowed us to distinguish between core states and those dynamically responding to the environment and depicted their adaptive potential. Cell states represent an essential trait that is robust to genetic perturbations but dynamic to environmental perturbations.

## Single-cell genotype-to-phenotype analysis highlights regulators of yeast cell states

Our atlas provides a comprehensive genotype-phenotype information at organismal level. To understand the genetic determinants of cell states, we assessed the enrichment or depletion of specific genotypes over the defined cell states (Supplementary Data 6,7). The wild type and most mutants (approx. 90%) were not biased towards any cell state in either condition tested (Fig. 4a,b and Supplementary Fig. 4a,b). It is worth noting that approximately 10% of mutants were significantly enriched in or depleted from specific cell states (253 in control and 331

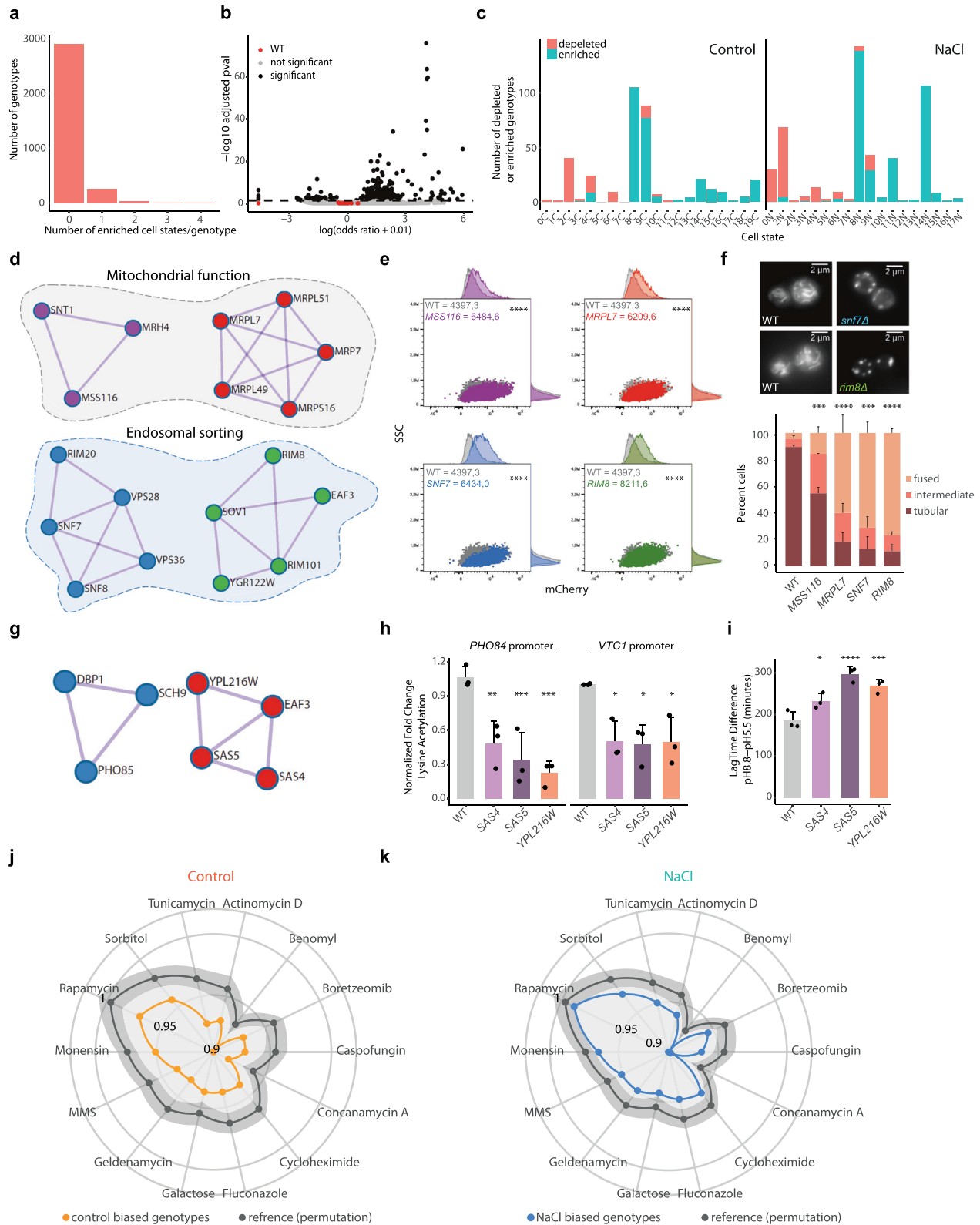

in stress; with 162 of them common to both conditions) (odds ratio >1, *p-value* < 0.05) (Supplementary Fig. 4c,d). Cell-state biased mutants tended to be enriched in or depleted of a single state and rarely in multiple states (Fig. 4a, Supplementary Fig. 4a) a pattern that is clone independent (Supplementary Fig. 4e,f). Surprisingly, cell state enrichment occurred more frequently than cell state depletion (Fig. 4b, Supplementary Fig. 4b), suggesting that there is a preferential role for preserving cell state plasticity. Our results suggest that cell

state organization is a highly robust biological process resistant to most genetic or environmental perturbations.

We then analyzed the distribution of the biased mutants across cell states. Two related states were observed to be preferentially enriched by mutants in control and stress conditions (8 C and 9 C in control and 8 N and 14 N in stress) (Fig. 4c and Supplementary Figs. 4g,h). Indeed, cluster 8 C (related to 8 N) displayed the aged iron-regulon signature, and mutants that impair mitochondrial functions

**Fig. 4 | Single-cell genotype-to-phenotype analyzes highlights regulators of yeast cell states. a** Distribution of the number of enriched cell states per genotype in control conditions. **b** Volcano plot shows the cell state enrichment of each mutant in control conditions (wild type is shown in red). Black line indicates the threshold for statistical significance (two-sided Fisher test Benjamini-Hochberg adjusted *p-value* < 0.05). **c** Number of genotypes enriched (blue) or depleted (red) per each cell state in each condition. **d** Protein enrichment network of mutants in control condition enriched in cluster 8 C (odds ratio >1, two-sided Fisher test Benjamini–Hochberg adjusted *p-value* < 0.05). Nodes are colored according to module enrichment (MDCE, see Methods). **e** Expression of the pFIT3 reporter (8 C) measured by flow cytometry, reporter expression is shown in the x axis against the side scatter 8SSC) for the indicated strains colored as in (**d**). Histograms represent density of the x and y axis. Graphs shows a representative experiment. Mean, standard deviation and significance to the wild type (paired t.test) (*n* = 3). **f** Representative images of mitochondrial morphology stained by MitoTracker for the indicated strains. Barplots show average of each mitochondrial morphology under control conditions (*n* = 3, 150 cells/strain). Paired t.test was performed comparing the frequency of tubular morphology between the wild type and mutant strains (*pvalues* 0,00019 (mss116), 6,42E-05 (mrpl7), 1,30E-05 (snf7), 0,00012 (rim8)). **g** Protein–protein interaction for genotypes enriched in cell state 9 C colored according to their MDCE. **h** Levels of Lysine acetylation determined by ChIP at the indicated strains and promoters in control conditions for the indicated promoters. Bars represent the mean and standard deviation of three independent biological replicates (dots). Benjamini-Hochberg corrected paired t.test is shown respect to the wild type (*pvalues* for the indicated strains (left to right) *PHO84*; 0.00219, 0.00043, 0.00022, 7.6e-05; *VTC1*; 0.0126, 0.0126, 0.0126, 0.0097. **i** Growth curve of the indicated strains in control (CSM pH=5.5) or alkaline conditions (CSM pH=8.8). The bars represent the average and standard deviation between the difference in Lag time between the alkaline media respect to control (*n* = 3). Benjamini–Hochberg corrected paired t-test is shown respect to the wild type (*p-value* for the indicated strains (left to right) 0.0171, 0.0002, 0.0012, 0.0082. **j** Radar plot shows the fitness score across a stressor panel for mutants enriched in specific cell states in control conditions (orange) or a random permutation as a reference (gray). Light gray ribbons show the 95% confidence interval. **k** Fitness score as in **j** for mutants enriched in cell states in stress conditions (blue). Source data are provided as a Source Data file.

(mitochondrial translation and ATP transport) and iron homeostasis accumulated in these clusters and displayed this iron-dependent premature aged signature in both conditions (Fig. 4d). Of note, mutants defective for endocytosis (i.e., endosomal sorting and ESCRT complexes) also accumulated in the same state (Fig. 4d). These findings suggest, and reinforce the initial observations, that defects in mitochondrial function lead to impaired iron homeostasis[29] promoting a premature aged transcriptome as the most common phenotype induced by loss-of-function mutations.

We validated that indeed representative mutants (based on the protein interaction network) of the mitochondrial function or endosomal sorting (8 C and 8 N) showed a higher expression of a reference marker for the iron-starvation cluster (pFIT3 mCherry) when compared to wild type (Fig. 4e). The expression of the iron-cluster leads to an aged phenotype (Fig. 2e,f), hence we examined the mitochondrial morphology in these mutants. While mutants related to mitochondrial function (mss16 and mrpl7) displayed an increased percentage of cells with fused mitochondria, mutants related to endocytosis (snf7 and rim8) showed even a more severe phenotype with most of the population (>60% of cells) displaying fused mitochondria (Fig. 4f). These results are in agreement with the described role of Snf7, a member of the ESCRT complex, in macromitophagy[33]. Our results demonstrate that gene deletions can lead to the accumulation of cells in specific cell states, characterized by distinct expression patterns. Although most cell states are shared across genotypes, mutations influence their relative abundance. This information allows us to leverage the similarity in transcriptional phenotypes to link functionally related genes and elucidate phenotype-genotype interactions.

The transcriptional response to genetic perturbations also provides information-rich profiles to naturally infer and predict gene function. For example, in the control condition, the second cell state with the larger number of biased mutants (9 C) showed a strong transcriptional signature similar to phosphate starvation (expression of the PHO pathway and VTC complex (Fig. 3b)[34]. Accordingly, genotypes depleted from this cluster include hmt1, a mutant in an arginine methyltransferase and known positive regulator of phosphate genes[35], whereas genotypes that accumulated in this cluster include mutants in known negative regulators of phosphate metabolism (e.g pho85, kcs1)[36,37] (Fig. 4g and Supplementary Fig. 4e). Additionally, we identified other mutants involved in histone acetylation, namely the SAS (sas4, sas5 a trimeric histone acetyl transferase complex) and Nua4 (eaf3) complexes, as mutants with increased expression of the phosphate genes and biased towards cluster 9 C, suggesting a role for these complexes in phosphate homeostasis (Fig. 4g).

Within the protein interaction network that contained the SAS and NuA4 complexes, we identified *YPL216W*, a paralog of unknown function of *ITC1*, a component of the Isw2 chromatin remodeling complex involved in gene silencing. Given its link with histone acetylation complexes, we sought to validate its involvement in the regulation of the phosphate regulon by assessing the levels of acetylation at phosphate genes by chromatin immunoprecipitation (ChIP). Indeed, the deletion of *YPL216W*, *SAS4* and *SAS5* led to a decrease in lysine acetylation at promoters of phosphate genes (i.e., *PHO84* and *VTC1*) (Fig. 4h). It is known that alkalization of the media resembles phosphate starvation and mutants defective in phosphate metabolism like *pho85* (Supplementary Fig. 4e) displayed reduced growth in alkaline media. Based on the phosphate-starvation transcriptome of *ypl216w* and *sas4 and sas5* deletions, we hypothesized that these mutants should also be sensitive to alkaline conditions. As expected, these mutants showed a significant slower growth as observed by the lag time difference when we monitored growth in standard rich (pH=5.5) and alkaline media (pH=8.8) (Fig. 4i), validating a role for these genes in the regulation of phosphate homeostasis.

To determine the overall phenotypic effects of cell state confinement, we evaluated the fitness of all mutants with a biased cell state distribution (253 from control and 331 from stress, Supplementary Fig. 4c) using available environmental screens across 14 stressors[38]. Cell state-biased mutants from both conditions systematically displayed lower fitness scores compared to a random permutation, regardless of the stressor type (Fig. 4j, k). These observations indicate that cell state variety is a defining property of healthy populations, and that mutants showing enrichment towards certain cell states associate with reduced fitness.

Therefore, by integrating single-cell transcriptome data, cellular state mapping, and genetic perturbations, by using the cell state distribution, we identified underlying mutations that promote cell state attraction or deplete cells from a state. The Yeast Transcriptome Atlas provides a unique resource that links genotype-transcriptome-phenotype, offering insights into gene function that are difficult to discern at the population level.

## Different cellular functions drive transcriptional heterogeneity under control and stress conditions

Transcriptional heterogeneity is a source of cell plasticity, and it has an impact on cell phenotype. Leveraging on the single-cell resolution of the Perturb-seq, we reasoned that scoring the degree of transcriptional heterogeneity shown in each mutant would reveal genetic drivers of heterogeneity and their conservation across conditions. By applying an SVD-based leverage score as previously reported genome-scale human Perturb-seq[11], we determined the genetic perturbations that

resulted in significantly deviated gene expression from the wild type. Of note, the average leverage score strongly correlated across conditions (Supplementary Fig. 5a). While the variance of the leverage did not linearly correlate with cell number, lower cell counts were associated with higher variability in leverage scores (Supplementary Fig. 5b,c). We then used the standard deviation of the scaled leverage score to identify mutants with increased (negative regulators) or decreased (positive regulators) heterogeneity, using a 30% threshold relative to the wild type (see Methods). While most mutants, in both conditions, did not lead to changes in transcriptional heterogeneity, we identified a larger fraction of negative regulators of transcriptional heterogeneity (approx. 150 mutants) than positive regulators (approx. 20 mutants) (Fig. 5a,b) (Supplementary Data 8). This pattern mirrors the observations in the human Perturb-seq dataset[11] (Supplementary Fig. 5d,e), thereby suggesting that transcriptional heterogeneity is kept within a defined range and only a few mutations alter it.

To understand the nature of the potential drivers of heterogeneity, we projected the negative regulators of each condition individually onto the yeast genetic interaction map[39,40]. Negative drivers spanned a variety of functions in both conditions. As expected, under control conditions, chromatin, transcription, and translation mutants were present, but surprisingly, genes related to vesicle trafficking, cytokinesis, mitochondrial function, and tRNA modifications were also included (Fig. 5c). Of note, mutants related to mitochondrial dysfunction generate increased heterogeneity in mammals[11]. Upon stress, we found that negative regulators involved some specific cellular functions, such as peroxisome function, nuclear transport, and chromatin/transport. However, mutants related to mitochondrial function were also among the most represented drivers (Fig. 5d). While both control and stress drivers showed shared functions, such as mitochondrial function, the identity of these genes poorly overlapped (10%, Fig. 5e and Supplementary Fig. 5f,g). Surprisingly, transcriptional heterogeneity is regulated by several core processes, and it is condition-specific, suggesting that the identification of molecular drivers should be performed for each specific condition of study. Therefore, our extensive genome coverage provides a comprehensive annotation of genes and pathways for their role in transcriptional heterogeneity.

To assess the physiological impact of transcriptional heterogeneity, we extracted cell fitness scores from the top and bottom 50 drivers for each condition from 14 phenotypic screens, as shown in Fig. 4j,k[38]. Mutants with decreased heterogeneity under control conditions showed significantly higher fitness upon stress than genotypes with high variability[41] (Fig. 5f). Conversely, mutants with high transcriptional heterogeneity in osmostress displayed consistently lower fitness in most stress conditions (Fig. 5g). These observations suggest that transcriptional heterogeneity is a quantitative trait that influences cell fitness. Our results support the notion that excessive transcriptional heterogeneity generates deviated transcriptional patterns from the wild type that ultimately render cells vulnerable to stressors, thereby weakening cell adaptability.

## Discussion

Functional profiling of the eukaryotic genome was pioneered in yeast through the use of the YKOC[38,39,42–46] and has systematically been applied by many groups at the phenotypic level (Yeast Phenome[16]). However, the YKOC is not suitable for single-cell studies. Here we performed a single-cell genome-scale Perturb-seq in *S. cerevisiae* by modifying the initial structure of the (non-essential) YKOC to generate RNA barcoded mutants, which allowed us to exploit the genotype-transcriptional phenotype, including clonal resolution. Thus, our study provides an updated version of the YKOC that contains genotype RNA barcoding structure suitable for genome-scale Perturb-seq. Our 3'UTR-based barcoding deletion strategy systematically profile isogenic perturbations with clonal resolution which can be reassayed at any time and condition as a full collection or for a subset of mutants. Of

note, unlike CRISPRi based approaches, deletion-based strategies are permanent perturbations in isogenic cells. However, integrated deletions do not allow to profile essential genes and may result in compensatory mechanisms. Our design extends the functionality of the YKOC providing a framework for tracing clonal expression dynamics by scRNA-seq as well as integrating a way to trace clonal fitness in conventional phenotypic screens.

The analysis of the library under two different conditions enabled us to generate a genome-scale Yeast single-cell Genotype-to-Transcriptome Atlas that covers most of the genome of a eukaryotic organism (more than 75% of the genes) with single cell resolution and in two different conditions recapitulating previously reported transcription patterns obtained in bulk[21]. Therefore, the breadth and depth of the atlas provides a high resolution transcriptional data to integrate with the vast amount of information acquired over decades and will serve to assess transcriptional single cell data for ubiquitous studies. In our study we provide evidence for multiple uses of the atlas. An interesting finding was the identification of shared core transcriptional states within a cell population. While many perturbations induced changes in expression, only few mutants dramatically changed the transcriptional landscape perhaps due to redundancy in gene function. However, about 90% of the mutants retained cell state plasticity, highlighting the robustness of cell states against genetic and environmental perturbations.

We categorized cell states into "responsive", which are modulated by external conditions and promote specialized adaptive functions, and "core", which are genotype- and condition-independent. These transcriptional states involve developmental and core functional transcriptional programs (oxidative phosphorylation, cell cycle, cell wall morphogenesis, protein homeostasis, and stress responses, among others). Interestingly, some of the core states associated with protein homeostasis, cell cycle, oxidative phosphorylation and stress response are also found in human Perturb-seq[11], thus indicating partial conservation of functional cell states. The analyzes of the transcriptional programs in each state served to deconvolute different degrees of phenotypical development or distinct transcriptional paths that lead to aging phenotypes.

The high amount of information provided by this Perturb-seq yields a comprehensive snapshot of cellular states as a function of genotype and condition. We harnessed the potential of 10% of the mutants that showed biased cell state accumulation. Some mutations irreversibly confined cells to specific transcriptional states, acting as stage attractors, whilst others, depending on the environmental conditions, relocated or pushed cells into specific states. Notably, mutations in mitochondrial homeostasis emerged as a core function leading to cell state rigidity, an event that promotes aging through iron-starvation like signatures. The transcriptional response of the mutants also provides information to infer and predict gene networks and functions. For example, in the control condition, the second cell state with the larger number of biased mutants displayed a strong transcriptional signature related to phosphate metabolism and this served to identify regulatory layers of phosphate homeostasis.

In addition to uncovering the genetic underpinning of cell states, the catalog of transcriptional phenotypes captures genotype-specific gene expression patterns, thus enabling the identification of positive and negative regulators of transcriptional heterogeneity. Whereas in higher eukaryotes these have been linked mainly to changes in copy number, our data in yeast suggest that negative regulators involve several cellular functions, including transcription, translation, and metabolic functions. Of note, some of the most abundant negative regulators involved several mitochondrial functions. Similarly, perturbation of mitochondrial dysfunction in the human Perturb-seq leads to increased heterogeneity[11]. These results highlight the importance of considering both global and condition-specific effects when assessing the impact and regulation of transcriptional heterogeneity.

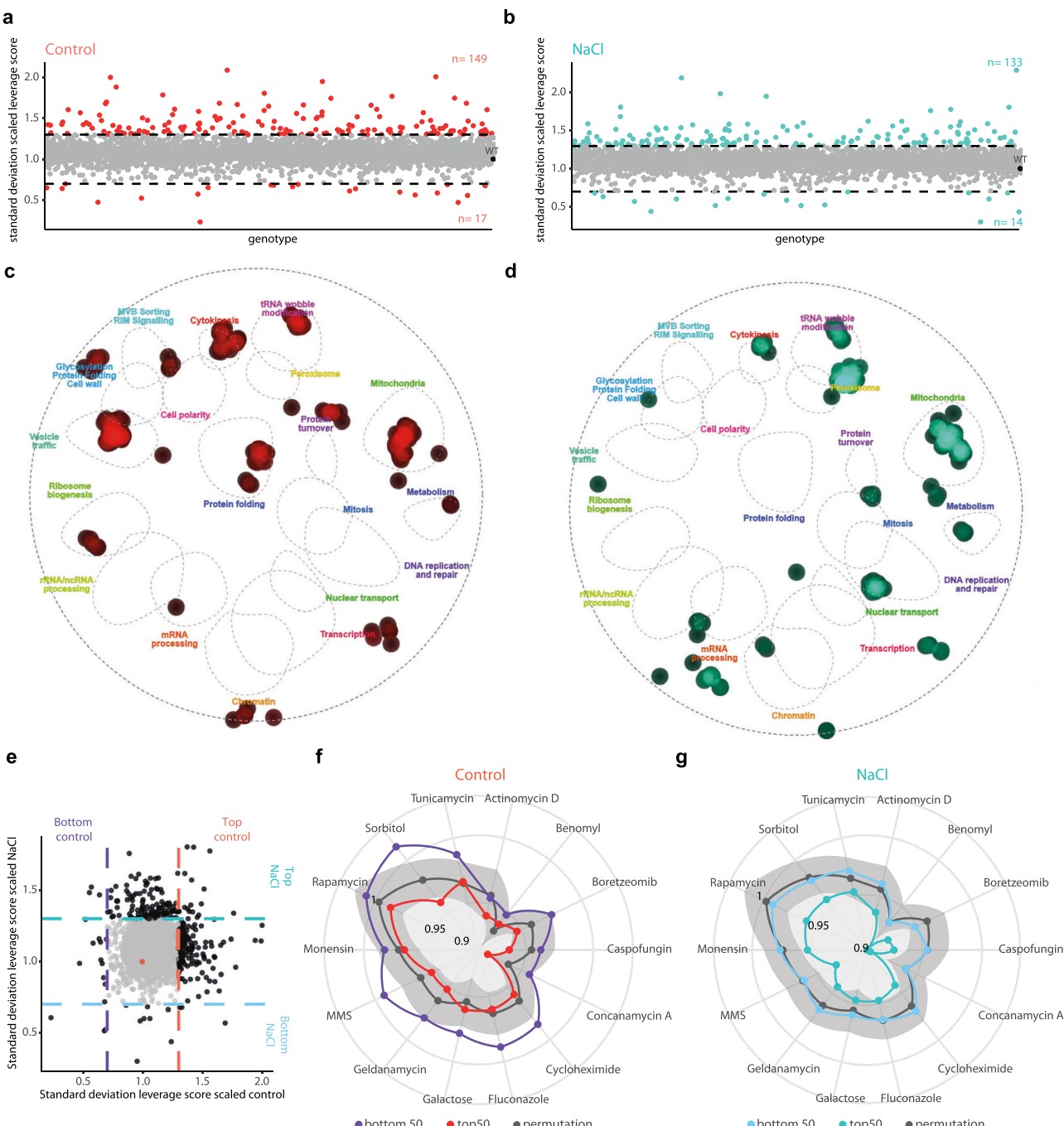

**Fig. 5 | Different cellular functions drive transcriptional heterogeneity under control and stress conditions. a,b** Distribution of the standard deviation of heterogeneity (scaled leverage score, y axis) per each genotype (x axis) in control **a** and stress **b**. Genotypes in the x axis are plotted in the same order. Dashed lines indicate the >30% increase or decrease threshold compared to the wild type. Colored points represent positive or negative regulators. The total number of genotypes above and below threshold is shown in graph. **c,d** Projection of negative regulators of heterogeneity per each condition (control, **c** and NaCl, **d**). The list of candidates overlaid with the yeast genetic interaction network to associate genes to function. Points represent the density of mutants over each indicated cellular function from the cell map. **e** Comparison of transcriptional heterogeneity per genotype (standard deviation scaled leverage score). Each point represents a genotype in control (x axis) or stress (y axis). Points are colored according to the >30% difference respect to the wild type (red point). **f,g** Radar plot shows the fitness score across a stressor panel for mutants of top50 (red and turquoise) and bottom 50 (purple and blue) genotypes with increased/decreased heterogeneity per each condition (**f** control and **g** NaCl) against a random permutation as a reference (gray). Light gray ribbons show the 95% confidence interval. Source data are provided as a Source Data file.

Predicting molecular responses upon any kind of perturbation at the single cell level is a core question in biology and a challenge that requires genome-wide perturbations[11,47,48]. We believe that our dataset provides an experimental framework with paired genetic and environmental perturbations, serving as a rich resource from in silico screening. In summary, the Yeast single-cell Genotype-to-Transcriptome Atlas provides a reference map, enabling both hypothesis-driven and hypothesis-generating exploration of cellular behaviors at several levels, such as genotype-phenotype relationships, and it can be used to identify conserved traits between eukaryotes.

## Methods

### Generation of a single cell transcription atlas by using a modified yeast knock out collection

To generate a genome-scale library of RNA-traceable deletion mutants we reengineered the deletion of the original yeast knockout collection (YKOC)[49]. Briefly, we generated a PCR cassette to replace the G418 resistance marker with *URA3* to shorten the heterologous terminator linking the original Downtag barcode to the 3'UTR of *URA3* to a minimum (43 nt) and added a clone barcode (5 random nucleotides) downstream of the *URA3* STOP codon (see below). This strategy allows the labeling and transcriptional tracing of genotypes and clones.

### Oligonucleotides used for the modification of the YKOC collection

Oligonucleotides for strain generation (Integrated DNA Technologies) were purified by PAGE purification and resuspended with nuclease free water (Thermo, 10977035) at a final concentration of 100 μM.

**For YKOC deletion strains.** *OMN761:* CACATCACATCCGAACATAAAC AACCatgggtaaggaatcgaaagctacatataaggaac

*OMN773:*TCGATGAATTCGAGCTCGTTTTCGACACTGGATGGCGG CGTTNNNNNttagttttgctggccgcatc

**Primers to generate WT strains.** *OMN774:*GATTCGGTAATCTCCGAG CAGAAGGAAGAACGAAGGAAGGAGCAGACATGGAGGCCCAGAATACC

*OMN775:*ATTTGTGAGTTTAGTATACATGCATTTACTTATAATAC AGTTCGGTGTCGGTCTCGTAGNNNNNNNNNNNNNNNNNNNNATC GATGAATTCGAGCTCGTTTTCGACACTGGATGGCGGCGTTNNNNN ttagttttgctggccgcatc

For *OMN773* and *OMN775* a stretch of 5 random nucleotides (N5) was added after the stop codon to include a clone barcode. Additionally, to generate barcoded WT strains, OMN775 contains an additional stretch of 20 nt.

The PCR product resulting from *OMN761-OMN773* contains homology to regions within the original deletion cassette targeting the junction of the p*TEF1*-KAN and the 1 nt upstream of the genotype barcode (D2-Downtag-D1), therefore shortening the terminator (from 262 nt in the original YKOC to 43 in the RNA-barcoded collection) enabling the use of the endogenous terminator. A total of 7 different WT strains were generated with distinct clone and genotype barcodes as controls. These wild type strains were verified by Sanger sequencing. For wild type strains the PCR product resulting from *OMN774-775* contains homology regions upstream and downstream of the URA3 loci.

Generation of the PCR cassette was done by pairing *OMN761-OMN773* and *OMN774-OMN775* for deletion strains and WT strains respectively. Expand High Fidelity PCR System (Roche, 11759078001) was used to amplify the *URA3* marker using 10 ng pRS406 as a template in reactions of 100 μl (Buffer# 2 10X with MgCl₂, dNTPs mix 1 mM (Promega, U1420), 1 μM OMN761 or OMN773 (Fw), 1 μM of OMN773 or OMN774, and 2.6 U (0.75 μl) of Expand High Fidelity Enzyme Mix. PCR product was purified using PB Buffer (Qiagen, 19066), transferred to a DNA purification column (EconoSpin Columns, 1910-050) and centrifuged for 1 minute at 21000 × *g*, washed once with PE Buffer (Quiagen, 19065) and eluted using nuclease free H₂O. Finally, the purified PCR product was diluted to 400 ng/μl.

### Sequence of the RNA-barcoded Yeast Knock Out deletion structure

GATGTCCACGAGGTCTCTNNNNNNNNNNNNNNNNNNNCGTACGCTGC AGGTCGACGGATCCCCGGGTTAATTAAGGCGCGCCAGATCTGTTT AGCTTGCCTCGTCCCCGCCGGGTCACCCGGCCAGCGACATGGAG GCCCAGAATACCCTCCTTGACAGTCTTGACGTGCGCAGCTCAGG GGCATGATGTGACTGTCGCCCGTACATTTAGCCCATACATCCCC ATGTATAATCATTTGCATCCATACATTTTGATGGCCGCACGGCGC

GAAGCAAAAATTACGGCTCCTCGCTGCAGACCTGCGAGCAGGGA AACGCTCCCCTCACAGACGCGTTGAATTGTCCCCACGCCGCGCC CCTGTAGAGAAATATAAAAGGTTAGGATTTGCCACTGAGGTTCTT CTTTCATATACTTCCTTTTAAAATCTTGCTAGGATACAGTTCTCAC ATCACATCCGAACATAAACAACC**atgggtaaggaa**tcgaaagctacata-taaggaacgtgctgctactcatcctagtcctgttgctgccaagctatttaa-tatcatgcacgaaaagcaaa-caaacttgtgtgcttcattggatgttcgtaccaccaaggaattactggagttagttgaagcat-taggtcccaaaatttgtttactaaaaacacatgtggatatcttgactgattttttccatggagggca-cagttaagccgctaaaggcattatccgccaagtacaattttttactcttcgaagaca-gaaaatttgctgacattggtaatacagtcaaattgcagtactctgcgggtgtatacagaatagca-gaatgggcagacattacgaatgca-cacggtgtggtgggcccaggtattgttagcggtttgaagcaggcggcggaagaagtaa-caaaggaacctagaggcctttttgatgttagca-gaattgtcatgcaagggctccctagctactggagaatatactaagggtactgttga-cattgcgaagagcgacaaagattttgttatcggctttattgctcaaagagacatgggtggaaga-gatgaaggttacgattggttgattatgacacccggtgtgggtttagatgacaaggga-gacgcattgggtcaacagtatagaaccgtggatgatgtggtctctacaggatctga-cattattattgttggaagaggactatttgcaaagggaagggatgctaaggta-gagggtgaac**GTTACAG**aaaagcaggctgggaagcatatttgagaa-gatgcggccagcaaaactaa**NNNNN**NAACGCCGCCATCCAGTGTCGAAAA<u>C GAGCTCGAATTCATCGAT</u>NNNNNNNNNNNNNNNNNNNN<u>CTACGA GACCGACACCG</u>

Sequence legend:

N: Random nucleotides N20 genotype barcodes from original YKOC and N5 represents the added clone barcode.

Lowercase: open reading frame of the URA3 gene. Bold letters denote nucleotides remaining from the original G418 resistance. These nucleotides were kept to increase the efficiency of the integration.

Underlined regions: original D2 and D1 sequences from the original YKOC

### Modification of the yeast knock out collection

For the generation of the modified strains, frozen glycerol stocks from the haploid yeast knock out collection were grown on YPD (Yeast Peptone Dextrose medium) supplemented with G418 (Geneticin, 200 mg/L) into 96 well plates using a Robot (Singer instruments). Strains were allowed to grow for 48 h until saturation at 25 °C before transformation.

For high-throughput liquid transformation, 10 μl of saturated cultures were transferred and diluted into deep 96 well plates containing 700 μl of YPD using EpMotion 96 (Eppendorf). Cells were allowed to grow for 6 h, media removed and 200 μl of LiAc solution was dispensed using a Multidrop Combi (Thermo, 5840300). A 1:1 mix of 40 μl of ssDNA (10 mg/ml) and a denatured PCR cassette for deletion strains (PCR product OMN761-773) and for WT strain (OMN774-OMN775) (400 ng/μl) was added to the cell-LiAC mix with an EpMotion. Last, 300 μl of 50% PEG solution was added to the mix. Yeast transformation plates were incubated at 30 °C for 1 h and after 35 μl of DMSO was dispensed and briefly shacked with Multidrop Combi. Transformations were heat shocked for 30 min at 42 °C using a water bath, cells were centrifuged, and transformation reagents removed. Pelleted cells were washed with 200 drop out media (CSM URA, MPBio, 1145112-CF) centrifuged, resuspended in 300 μl of URA- and incubated at 30 °C overnight. Then plates were centrifuged, and cells resuspended in fresh URA- media for 48 h. Transformation efficiency was determined by transferring 10 μl of the transformation into fresh URA- media and check for growth using a microplate reader (Synergy HX1, Agilent Technologies). Grown cells were mixed with URA-50% glycerol to make frozen stocks of the RNA-barcoded Yeast Knockout Collection. The genotype genomic location of reengineered strains was validated using the 3'UTR region of *URA3* and the median distance to the expected deletion has can be found in Supplementary Data 1.

## Yeast Growth and harvest for the Perturb-seq experiments

Frozen glycerol stocks of each individual mutant from the transcriptionally barcoded yeast knock out collection were recovered on URA- media at 25 °C for 48 h until saturation. The next day, 5 µl of cultures were refreshed into another 96 well plate containing 200 µl of YPD. Cells were allowed to grow for 6 h until they reached mid exponential phase average $OD_{660}$ 0.6–0.8. To maximize representativeness of each mutations optical density for all plates was assessed after seeding and before pooling using a Synergy HX1 reader. Cells then were pooled together into large flasks and shacked vigorously to ensure a homogeneous mixture.

We subjected or not cell pools to osmostress (0.4 M NaCl for 15 min) and followed the cell fixation protocol from GEXSCOPE® Microbial Single Cell RNA Library Kit HD (Yeast) (4161031). The selection of this experimental conditions is based on a combination of extensive transcriptomics data[50,51] that determines the peak of expression of osmoresponsive genes. Briefly, 100 ml cells where centrifuged 1 min 850 × g, media poured off and cells immediately fixed and resuspended in 10 ml of ice cold 80% Methanol (Scharlab, ME0301005P). For each condition methanol-fixed cells were split into 20 different aliquots (1 ml) from each condition (control/NaCl) and stored at −20 °C, to avoid multiple freezing and thawing.

## Library generation

To perform the yeast, Perturb-seq experiments we followed GEXSCOPE® Microbial Single Cell RNA Library Kit HD instructions (4161042, Singleron Biotechnologies). Briefly, A methanol frozen aliquot of each condition was equilibrated for 15 min at 4 °C (ice) before pelleted (2 min 1640 g at 4 °C). Supernatant was discarded and cells were washed twice with 700 µl of rehydration buffer (DPBS1X, BSA 20 mg/ml (Thermo Fisher Scientific, 14190144 and AM2616), RNase inhibitor (40U/µl) (Takara, 2313 A), Actinomycin D (2 mg/ml) (Sigma, A1410-2MG)). To disaggregate potential cell clumps we filtered twice with PluriStainer (PluriSelect,43-50040-03) and cells were counted with a Neubauer chamber. A total of 220,000 cells (120 µl) of the yeast suspension per matrix-cartridge in the High Density Singleron Matrix v1.0.1instrument. For each batch, a total of 2 matrix-cartridges were run simultaneously each one loaded with control and NaCl simultaneously. Batch 11 was performed only with one stress sample. The position of each sample in the Singleron Matrix instrument (upper/lower slot) was exchanged per every run. A total of 29 matrix-cartridges were run (14 control and 15 NaCl).

Single-Yeast Partitioning, mRNA capture, reverse transcription, cDNA amplification and cDNA purification were performed according to manufacturer's instructions. For cDNA library amplification 8 PCR cycles were used. Samples were processed per run (as group of two samples, control/NaCl) until cDNA purification. The cDNA quality of each cDNA was assessed by Qubit (Q33231, Thermo Fisher Scientific, Qubit™ 1X dsDNA HS Assay Kits), the size and integrity of the full-length cDNA was inspected with a DNA Pico Bioanalyzer chip (Agilent technologies). Library preparation was done using 50 ng of purified cDNA as an input and library amplification was done with 10 PCR cycles. Finally, library quality check was performed with Qubit and library size was determined with DNA Pico Bioanalyzer chip. Equimolar pools of the 29 libraries were pooled together and sequenced in one S4 NovaSeq lane (CeGAT, Tübingen) using paired end 150 cycles (PEx150 cycles). A total of 3.58 Tb of data were generated from a single run.

## Targeted Amplification (TA-libraries)

For each of the 29 full length cDNA libraries, a total of 5 ng was used as an input to amplify the *URA3* transcript and the 3'UTR with a one-step PCR reaction. PCR was generated using the NEBNext Ultra II (E7645L, New England Biolabs) in a final volume of 25 µl following manufacturer's instructions. The primer design incorporates the P5/P7 sequences, the index barcode and the Illumina read 1/2 sequences

to enable direct sequencing of the purified PCR products. The resulting PCR products (102 bp excluding the primer overhangs) were purified using Ampure Beads at a 1.5× ratio and eluted with 30 µl of elution buffer. Library size was inspected by Bioanalyzer chip and equimolar amounts of each library were pooled together for sequencing using a NextSeq500 (Illumina).

## Read pre-processing, alignment, and filtering

To process the sequencing data (FASTQ), we used standard CeleScope (v1.14) pipeline from Singleron (https://github.com/singleron-RD/CeleScope). To process FATSQ files, we first trimmed the D1+downstream and D2+upstream reads originating from the knock-in loci to retain only the genotype sequences. Reads were aligned to a Yeast's sacCer3 reference genome with all the genotype sequences appended as additional contigs to obtain counts of gene and genotype barcode expressions. Sequences of the genotype barcode were downloaded from the Yeast Deletion Project (http://www-deletion.stanford.edu/YDPM/YDPM_index.html). Additionally, an artificial chromosome containing the clone barcodes (5 nucleotides), the common terminator region and the genotype barcode (D2-Downtag-D1) was added to the reference genome to enable genotype identification from the expression matrix. Example of an artificial genotype chromosome:

>bc-Systematic Name
NNNNNAACGCCGCCATCCAGTGTCGAAAACGAGCTCGAATTCATCGATNNNNNNNNNNNNNNNNNNNNNNNCTACGAGACCGACACCG

Naming of replacement strains; the original YKOC and commercial collections contain some replacement strains which represent repetitions for conflictive strains. The names of these replacement strains and genotype barcodes are identical between repetitions. To include these strains in the reference genome and to avoid a naming conflict, chromosomes of each repeated strain was named sequentially: bc-systematic name, bc-systematic name-2, or bc-systematic name-3.

## Analysis of the targeted amplicon library

Barcode and UMI information were extracted from FastQ files with Singleron Celescope software version 1.14.1, using commands 'celescope rna sample --chemistry auto' and 'celescope rna barcode', and Singleron v3 whitelist and linker files.

Resulting fastq files were imported into R. Clone information was extracted from the sequenced read at positions 21–25. In order to find the genotype information, the region between the D1 and D2 inserts was extracted by matching the corresponding sequences allowing at most 3 mismatches. Only reads containing both D1 and D2 were kept for further analysis. In case of multiple matches of any of the insert sequences the one with maximum start position was chosen. Reads were further filtered for starting matching positions of D1 between 85 and 90, and D2 between 48 and 51. The inserted sequence corresponding to knock out genotypes in the library were compared to the remaining sequences using the vmatchPattern R function with a maximum mismatch of 2. Cells assigned to more than one genotype with no genotype present in more than 70% of reads were discarded.

To combine both sets of assignments, cells with different genotypes assigned in both libraries were marked as "conflicted" and omitted from downstream analysis. Cells with the same genotype assigned in both libraries were given that genotype. Cells with genotype assigned in only one of the 2 libraries were also given the assigned genotype.

## Genotype position

Because the YKOC is a globally used resource the assessment of the location of each deletion has only been performed manually for selected mutants or only available for the homozygote diploid collection at a global scale with Whole Genome Sequencing[52]. Our strategy enables to define the genomic location of each deletion in the RNA-

barcoded YKOC collection by leveraging the high resolution of the targeted amplification which has coverage of the endogenous terminator. To assign the genomic position all the FASTQ reads were combined and aligned against the yeast genome (saccer3, SGD). The median nucleotide distance between the mapped read and the annotated STOP codon was done to assess the genomic position of the intended deletion. Genotypes with greater distances greater than 300 bp from the genomic loci were considered incorrect (Supplementary Data 1).

### Downstream processing

The complete scRNA Yeast Genome Dataset includes a single processed Seurat object (Seurat v4)[53] that contains all cells profiled in both conditions and the corresponding metadata. Additionally, we generated an additional Seurat object corresponding to the control or stressed samples and the corresponding metadata available at E-MTAB-14004. As it has been done before and to ease the information to the community given the wide usage of the YKOC we have reported all the information for all genotypes detected and removed incorrect genotypes for detailed analysis and experimental validation.

To generate the Seurat Objects, the outputs of Celescope were used to generate the corresponding cell expression matrix either by combining both conditions or for each condition individually. To normalize gene expression across cells, we applied the SCTransform procedure. In order to mitigate the cell cycle effects, we used the Seurat function *CellCycleScoring* to calculate and classify cells in the corresponding cell cycle phase using expression of canonical phase specific genes previously used in scRNA-seq (see Supplementary Data 4 for cell cycle distribution per genotypes/conditions and Supplementary Data 9 for gene lists)[17]. Then we regressed out the cell cycle scores by supplying the cell cycle variable genes to the *vars.to.regress* argument. The preprocessed objects and code allow to explore the data before and after cell cycle normalization. The final log normalized results were used for all downstream analysis. We followed standard Seurat clustering guidelines with the following parameters[7]. We also identified highly variable genes using the FindVariableFeatures from the Seurat package with nfeatures =1. To perform cell clustering, first we performed a linear dimensional reduction using the "RunPCA" function from Seurat Package using PC1 and PC2. Visualization of gene loadings for the complete dataset was done by using the VizDimLoading function of PC1. To cluster cells we then applied the Seurat pipeline FindNeighbors (dims 1:14) and FindClusters (resolution =1). To visualize the UMAPs we used RunUMAP (dims 1:14).

### Clone comparison

Only the 220 genotypes with more than 200 cells were considered for the analysis. For each genotype, clones with more than 9 cells were compared against the rest of the clones (with at least 3 cells) using the FindMarkers function from Seurat (adjusted *p-value* < 0.05).

### Cell state markers

To extract cell state markers, we applied the differential expression function included in Seurat through FindAllMarkers for the complete dataset (both conditions) and each condition individually. Gene ontology enrichments of upregulated cell state markers were performed using Metascape v3.5.20230501[54] default parameters and we used S. *cerevisiae* as input and output specie. The enrichment terms per each input list was downloaded and appended to Supplementary Data 2 and 3.

To visualize the expression signatures per each condition individually, we generated lists of the all genes upregulated genes of each cluster using the FeaturePlot (order=TRUE). To visualize the co-expression of the aging signature we used the UCell package AddModuleScore_UCell aging genes defined by RNA-seq. The top 15 aging genes were retrieved from Patnaik et al. (https://pubmed.ncbi.

nlm.nih.gov/35858543/)[55], and used to generate an unbiased cell signature, or either a signature split in iron-containing or non-iron signature. The gene list of induced Environmental Response genes was obtained from published datasets[31,32] using the updated list from Gasch et al.[32] (https://sgd-prod-upload.s3.amazonaws.com/S000343511/ESR_clusters_UPDATED_2017.xlsx). Gene lists used for signatures are listed in Supplementary Data 9. For violin plots, dashed line indicates the global mean of each corresponding graph and black lines indicate the mean of each variable. The statistical significance was calculated using the *stat_compare_means* function from the ggpubr package.

### Conservation cell state enrichment across conditions

To calculate the degree of similarities between states, we calculated the correlation of expression between the upregulated cell state markers across clusters identified in each condition independently. To calculate the degree of similarities between states, a two-sided Fisher Test was performed comparing the common upregulated markers (Benjamini–Hochberg adjusted *p-value* < 0.05 and average log2 Fold Change > 0.25) between conditions cell states.

For the label transfer analysis, we employed the built-in function from Seurat. First, due to the similarity of cluster 14 C (basal-stress) to the entire NaCl dataset, the markers for this cluster (stress genes) were removed from both control and NaCl to prevent all stress states mapping to 14 C. Then renormalized using the same procedure applied to the original control samples. Subsequently, we used the *MapQuery* function to transfer annotations from the control to the stress sample. The normalized proportions of cells transitioning between control and stress clusters in both directions are reported in Supplementary Data 5. Core states were defined as those that share at least 40% of cells with a stress state. Of not cell state 14 C was manually included as core state.

### Cell state genotype enrichment

Cell state enrichment per genotype was done for genotypes with ≥6 cells to avoid biases due to cell number. To determine the enrichment degree a two-sided Fisher Test was performed and the odds ratio of each genotype per cluster was calculated. We considered enriched genotypes if the odds ratio was >1 and adjusted *p-value* < 0.05 or as depleted if the odds ratio was <0 and adjusted *pvalue* < 0.05. Cell state enrichment per each genotype is reported in Supplementary Data 6 and 7 respectively.

To visualize the protein interaction between enriched genotypes in a specific cluster we used the Metascape v3.5.20230501 default parameters. These parameters only include physical interactions using STRING (physical score > 0.132) and BioGrid. Additionally Molecular Complex Detection (MCODE) algorithm was used to define densely connected networks within the Metascape default parameter app.

### Transcriptional heterogeneity

**Differential expression.** Differential gene expression was assessed using the Wilcoxon rank sum test. Each mutant was compared to wild type in control and stress samples separately. For each comparison, genes with 0 counts were first removed. For global transcriptome analysis (Fig. 1F-I and S1E-S1H) differentially expressed genes were considered with using a threshold log2 fold change ≥ 1 and ≤ −1 and *p-value* < 0.05 for upregulated and downregulated genes respectively. To optimize statistical power, independent filtering was applied based on mean normalized expression. To identify the optimal threshold for average counts, we: (1) Applied different thresholds to filter out genes with averaged counts less than the thresholds. (2) Performed *pvalue* adjustment using Benjamini-Hochberg procedure on the remaining genes that passed the threshold and (3) Counted the number of statistically significant genes with adjusted *p* values lower than 0.05. The threshold resulting in the highest number of significant genes was finally used. This threshold was found and applied separately for the control (<1.27) and stress treated samples (<1.19).

**Comparison with bulk datasets.** To compare the similarity between gene expression from our dataset to previously published bulk microarray samples reported by Kemmeren et al.[21] in control conditions. The expression table ([deleteome_all_mutants_controls.txt]) was obtained from https://deleteome.holstegelab.nl. For the genotypes and genes shared across both studies ($n = 837$), we compared the median number of genes with $p$-$value < 0.05$. Given that a comparable dataset does not exist for stress conditions, we scored the expression of the ESR signature (see Supplementary Data 9 for genes) across mutants for at least 6 cells.

**Leverage score.** The leverage score was calculated in the same way as in ref. 11, except that we used log normalized counts yielded from SCtransform procedure instead of plate level z-score because of the variable composition of the wild type cells in each plate that might introduce more variance. Specifically, we (1) Constructed a count matrix depicting cells in rows and genes in columns, consisting of all genes with mean expression >0.25 UMI counts per cell. (2) We calculated the top 20 left singular vectors for each plate using the partial SVD algorithm (scipy.sparse.linalg.svds) with the arpack solver and k = 20. Row wise square norm of the resulting $n$ by 20 matrices (where $n$ is the number of cells) was calculated to give the leverage scores. These scores were normalized so that the sum over all cells in each plate is 1. (3) To normalize leverage scores across plates, we log transformed the scores from the previous step, and z-score normalized these scores relative to the scores of wild type cells (subtracting the mean and dividing by the standard deviation of the wild type cells). We reported the leverage score and the scaled leverage score of each cell in the corresponding metadata. Supplementary Data 8 contain the average, standard deviation, variance of the leverage score raw and scaled per each condition.

We ranked genotypes according to the standard deviation of the leverage score (wild type =1). For both conditions, and similar to the cell state enrichment in Fig. 3, and only for genotypes with at least 6 cells. The standard deviation was calculated using base R functions and the dplyr package[56]. We then visualized the negative drivers (≥1.3 standard deviation leverage score) or positive drivers (≤0.7 standard deviation leverage score) per each condition. The projection into the yeast interaction network was done using TheCellMap using the overlay function with the default parameters (https://thecellmap.org/)[40]. Functional enrichment analysis of control or stress genotypes with increased (>30%) or decreased heterogeneity (<30% of the wild type strain) was analyzed the Spatial Analysis of Functional Enrichment (SAFE) build in TheCellMap, applying default settings.

**Experimental validations**

**Cell state reporters.** Recombinant DNA techniques and transformation of bacterial and yeast cells were performed using standard methods. To generate reporters for cell states, we used the MoClo Yeast Toolkit Modular cloning system[57]. Building of the plasmid constructs was achieved using Golden Gate assembly. Each reporter contains a transcription unit composed of: the corresponding promoter (700 bp upstream of the annotated ATG), UbiM degradation signal, florescent protein and terminator (300 bp downstream of annotated STOP codon). All cloned sequences were either mutated or synthesized to avoid of the BsmBI, BsaI, and NotI recognition sequences. Promoter and terminator sequences were amplified from BY4741 genomic DNA and purified using the MiniElute PCR purification (28004, Quiagen). Plasmids generated in this study are described in Supplementary Table 1.

Entry plasmids were generated using the MoClo guidelines and were amplified in *Escherichia coli* DH5α competent cells grown at 37 °C in LB medium supplemented with the corresponding antibiotic for selection. Plasmid extraction was done using the E.Z.N.A.® I Kit (D6942-

02, Avantor) and verified by Sanger sequencing. Purified plasmid was linearized with NotI (NEB) and integrated into the yeast genome. Standard yeast transformation was done using the LiAc method into the corresponding yeast background and colonies were selected by marker selection and colony PCR. Strains harboring cell state reporters generated in this study are described in Supplementary Table 2. Expression of the reporters was followed by flow cytometry (see below).

**Calcofluor white staining and mitochondrial morphology (bud scars; aging phenotype analyzes)**
Wild type cell or cells carrying the corresponding reporter, were grown to exponential phase in SC media. Cells were filtered with a 70 μm mesh and a total of 300,000 cells was sorted using a FACS Aria III cell sorter into 15 ml falcons. The top 2% of the population (*pFIT3*-mCherry-*tFIT3* and *pHUG1*-mCherry-*tFIT3*) were fixed by directly sorting into Ethanol 100%. Additionally, a random sort for the entire population or for a wild type strain were collected as controls using the same fixation strategy. Fixed cells were stored at 4°C until Calcofluor white staining.

To visualize and count bud scars, we stained cells with 200 μg/ml Calcofluor White Stain as reported[58] (18909 Fluka Analytica) with minor modifications. Briefly, the indicated populations of cells were sorted directly into Ethanol fixed cells and washed once with DPBS 1X to remove excess Calcofluor. Scars per each cell for each corresponding population were counted using at 100x magnification (Plan Apo VC 60x Oil objective) using a Nikon Eclipse Ti inverted microscope and an ORCA digital camera (Hamamatsu). Per each biological replicate a total of 250–300 cells were counted.

To assess mitochondrial morphology, we used the same procedure as above except that cells were sorted into rich media (YPD) containing 100 nM MitoTracker™ Red CMXRos (Invitrogen, M7512). Cells were incubated in this media for 30 min, washed with media without MitoTracker and fixed in YPD 4% formaldehyde and stored for imaging. The morphology of mitochondria was assessed for at least 100 cells per each biological replicate of the indicated strains. Representative images of mitochondrial morphology were selected and processed individually with Image J, the brightness and contrast of each cell has been individually modified to improve clarity.

**Chromatin Immunoprecipitation**
Cells were grown to mid exponential log phase and 50 ml were harvested at OD660 0.6. Fixation was done by adding 1% formaldehyde (Sigma, F1635) for 20 min and quenched with Glycine 125 mM for 15 min at room temperature. Cells were pelleted by centrifugation and washed with TBS 1X four times at 4°C. Chromatin immunoprecipitation was done as described in ref. 59 Briefly, a total of 0.5 μg antibody per sample against Lysine acetylation (Cell Signaling, 9441S) and conjugated to 25 μ rabbit Dynabeads® M-280 Sheep Anti-Rabbit IgG(Life Technologies, 11204D) overnight at 4 °C. Levels of Lysine acetylation was determined by qPCR using primers specific to the indicated promoters to: PHO84 (*Fw: GGACGTGTTATTTCCAGCAC and Rv: CAGGCAAAACGGGAGAAGAG*) and VTC1 (*Fw: TTGGCATCGC TATTTTCGGA and Rv: ACCGACCGTAACAAGCGATA*) and as loading control an intergenic region of the right arm of chromosome VI was used (*Fw: ACCACTCAAAGAGAAATTTACTGGAAGA and Rv: CTCGTTAG GATCACGTTCGAATC*). For qPCR Power SYBR™ Green PCR Master Mix (Life Technologies, 4368708) was used in a final volume of 10 μl following manufacturer's protocol and qPCR reaction conditions. The resulting Ct values each gene was normalized to the loading control and delta Ct method[60] as used to calculated abundance of acetylated Lysines. The values for the wild type strain set to 1 and used as a reference.

## Competition assays

Wild type strains carrying the indicated expression reporters were grown to mid exponential log phase and sorted using Aria SORP (Becton Dikinson). A total of 20,000 cells of each top 2% of the population was sorted and 20,000 cells of a wild type strain carrying a constitutive GFP (*pTEF1*) was sorted on top in a final volume of 200 µl of rich media (YPD). A total of 150 µl of the mixed culture was fixed with 4% formaldehyde as time 0. The remaining culture was evenly split into YPD and YPD 1 M NaCl. Cells were diluted every 24 h and fixed after 48 h.

## Flow cytometry analysis

Cells were recorded from each sample according to their FSC and SSC distributions and unmixed to identify the fluorescence signal for each fluorophore (mCherry, iRFP or GFP). For competition assays, cells were gated based on the constitutive expression of GFP of the wild type strain versus the side scatter for three biological replicates. To read the expression of pFIT3-mCherry reporter in mutants in mutants enriched in cluster 8 C, the full spectrum of 10,000 cells were recorded Cytek® Aurora (4-laser and 64 Fluorescence Emission Detection Channels) gated according to the FCS and SSC distributions. The unmixed signal was used to assess the expression distribution of each mutant against the wild type and the mean expression (arbitrary units) per each strain and biological triplicates. The mean and median expression of each strain was obtained for 3 biological replicates. Cytometry data were analyzed using FlowJo™ Software (BD Life Sciences). A representative example of the FACS gating strategy for sorting and analysis is shown in Supplementary Fig. 2i. For all experiments the first gate was performed using the Forwards Light Scatter (FSC-A) side scatter (SSC-A) to select for cells. For isolation of the top 2% populations the first gate was subsequently used to isolate single cells using the FSC-A (area) and FSC-W (width). Single cell particles (second gate) were then used to isolate top 2% expression population using FSC-A and the reporter of interest (mCherry or iRFP). For competition assays and expression analysis, cells were gated using the same strategy (FSC-A and SSC-A) and the expression of the gene of interest or population of interest (GFP-mCherry, GFP-iRFP) was assessed from this gate.

## Comparison of fitness scores

This analysis was performed using Supplementary Table 1 (Mutant Fitness Conditions) from https://pubmed.ncbi.nlm.nih.gov/33958448/, encompassing 14 stressing conditions and 4429 genotypes. Only common genotypes between the publicly available dataset and our data were included. Three gene sets were built: the first comprised the top 50 genotypes with the highest standard deviation of the scaled leverage score, the second consisted of the bottom 50 genotypes with the lowest standard deviation of the scaled leverage score, and the third, called biased, that comprised enriched genotypes (Benjamini–Hochberg adjusted *p-value* < 0.05 and an odds ratio > 1) identified through the Fisher test. A one-sided permutation test was conducted for each stress score and each gene set by comparing the mean of the gene set with the mean of 1000 randomly selected unclassified genotypes of equal length.

## Growth curves

The indicated strains were grown below $OD_{(660)} = 1$ in rich media. For the experiment cells were washed three times with complete synthetic media pH=5.5 or pH=8.8. Then cells were diluted to $OD_{(660)} = 0.05$ in a final volume of 200 µl in a 96 well plate. Plates were incubated under orbital shaking at 30 °C in a Synergy H1 (BioTek® Instruments) and $OD_{(660)}$ was recorded every 30 min for 48 h. The Lag Time was calculated using the Gene 5 software (BioTek® Instruments).

## Reporting summary

Further information on research design is available in the Nature Portfolio Reporting Summary linked to this article.

## Data availability

The raw sequencing data for transcriptome and targeted amplification generate in this study are available in the Array Express database under accession code E-MTAB-14004. Fully processed Seurat objects containing the individual or combined datasets are available on Zenodo [https://doi.org/10.5281/zenodo.14062629]. Other data used in this manuscript: yeast fitness data across stressor panel were obtained from Costanzo et al.[38] (Supplementary Table 1). The leverage score from human Perturb-seq were obtained from Replogle et al.[11] (Table S2). The bulk expression table from Kemmeren et al.[21] was obtained from the authors repository https://deleteome.holstegelab.nl. Source data are provided with this paper.

## Code availability

All the code used in this study is available through Zenodo [https://doi.org/10.5281/zenodo.14062629].

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

## Acknowledgements

We thank the partnership with Singleron Singleron Biotechnologies GmbH for generously lending the Matrix equipment, the Singleron Bioinformatic service (Jaren Sia and Stacy Xu) for analysis and the Singleron R&D (Julie Laliberte) for technical support. We also would like to thank Dr. Cayetano González (IRB Barcelona), the IRB Functional Genomics Core Facility, the Flow Cytometry Unit from the Scientific and Technological centers (CCiTUB) and staff members Jaume Comas and

Ricard Alvarez, as well as Oscar Reina and Lidia Mateo (Biostatistics Unit IRB Barcelona) for technical support. We also thank Sandra Clauder and Dr. Cosimo Jan (from Dr. Steinmetz group) for generously sharing the original Yeast Knock Out collection. Additionally, we thank Aitor González, Dr. Pablo Latorre, Dr. Rodriguez-Fraticelli, Dr. Holger Heyn and Dr. Ivo Gut for helpful discussions during the initial conceptualization of the project. We would like to thank the Associación Española Contra el Cancer for supporting ADV through the AECC Excelencia program. Finally, we would like to thank Aida Fernández for technical support and special thanks to Mònica Romo for instrumental assistance in conducting the experimental validations. This work was founded by MICIU/AEI /10.13039/501100011033 and ERDF/EU to FP and EdN (PID2021-124723NB-C21/C22). Funding from the Ministry of Science, Innovation and Universities through the Centres of Excellence Severo Ochoa Award, and from the CERCA Programme of the Government of Catalonia and the Unidad de Excelencia María de Maeztu, funded by the AEI (CEX2018-000792-M). The Ramon y Cajal Program (Spanish Ministry of Science, RYC2021-033520-I) and La Caixa Junior (LCF/BQ/PR20/11770001) awarded to MNR. FP and EdeN are recipients of an ICREA Acadèmia award (Government of Catalonia).

## Author contributions

MNR, CS, LS, EdN, FP conceptualized the study. MNR, CS, AD, CS-O, YM performed the experiments and analyzed the data. MNR, EdN, FP acquired funding. EdN and FP supervised the project. MNR, CS-O, LS, EdN, FP wrote, reviewed and edited the manuscript.

## Competing interests

The authors declare no competing interests.
