## [Transparent Peer Review file · Nature Communications]

A single-cell resolved genotype-phenotype map using genome-wide genetic and environmental perturbations

Corresponding Author: Dr Francesc Posas

Version 0:

Reviewer comments:

Reviewer #2

(Remarks to the Author)

The authors engineered a new budding yeast knock-out library where deletion barcodes are expressed as RNA molecules and can therefore be detected by RNA sequencing. This is particularly useful as it opens the way to genetic screens with a read out based on single cell RNA sequencing (scRNA-seq). To showcase this resource the authors performed two experiments during rapid growth and in response to stress (0.4M NaCl, 15 mins). In addition to describing a great resource, this allowed to gain insight into cellular states plasticity and their reliance on genotypes. It also uncovered new knowledge on diverse genes function in single yeast cells as well as on the regulation of phenotypic heterogeneity. It is an interesting paper and I have listed a series of issues below that should be considered before publication.

Major:

1. Figure 1D: It would be useful to show projections of each axis as density histogrammes as on figure 4E and to report the median in addition to the mean.
2. Figure S1D: I find the label “clone markers” a bit confusing. These are the number of regulated genes compared to control and not number of clones correct? Shouldn't it read “number of regulated genes”? Also, is it really accurate to say that most mutants are in the first bin? Considering the tail the number of clones in the (0,1) bin and those in all other bins seems quite balanced. This need to be clarified.
3. Figure 1J-L: The authors state that the stress response genes are driving the split of the cells from the two conditions on the UMAP into distinct populations. How about a technical batch effect? The design with 29 libraries seems robust to this but it might be good to comment.
4. Figure 1L: “we could detect similar cell states within each condition...”. Does this figure really show that states are matching?
5. Are any of the states enriched for specific cell-cycle stages? This could be extracted from the scRNA-seq data and would help connecting this work with a large body of yeast literature.
6. Figure 2E-F: I believe the control population here is problematic. Top 2% expressing cells for aging markers are compared to randomly sorted cells. Levels of gene expression correlate globally with features like cell size or metabolic status. To take this into account the control should be the 2% higher expressors of the constitutive reporter and not a random set.
7. Figure 3: The use of “intrinsic” and “extrinsic” is a bit confusing. Some states could result from molecules secreted by cells within the culture. Such states would result from responses to external molecules as with the NaCl treatment but in this case would be called intrinsic. Do I make sense? This needs clarification.
8. Figure 3B: “For example, under stress, markers of the hypoxic aging and phosphate metabolism clusters (4C-15C and 9C, respectively) were widely expressed...”. This is possibly due to image quality, but to me it seems that those markers disappear upon stress induction instead. This needs clarification.
9. Figure 3C: On the figure, states have most of the time more than one counterpart in the other condition. Doesn't it challenge the statement that states are shared between conditions? To me it suggests that states are mostly different between control and stress with partial overlap. This needs clarification.

10. Figure 3F: Same comments as in point 5 applies here.

11. P9: "...is independent of cell number (Extended Data Fig. S5B. S5C)". I am not sure I see this on the figure. To me it looks like the scores ≥ 1.4 (used as cut off on 5A and B) are clustered on the left of the graph in strains with lower number of cells (< 64 cells). Am I missing something? If not, more validation is required to ensure that higher scores observed in certain mutants are not a result cell number and that biological conclusions are supported by the data.

Minor:

1. Figure S1B: Should one of the D2 labels be D1?

2. P7: "Our results hence indicate that phenotypes can be promoted by alterations in gene function and highlight the potential of using transcriptional phenotypes to define novel phenotype-genotype interactions." As it is, this statement reads a bit trivial? Maybe rephrase?

3. P.8: "we have defined the regulatory logic that governs cell state". This is a very strong statement. What do the authors mean here exactly?

(Remarks on code availability)

Reviewer #3

(Remarks to the Author)

In this paper Nade-Ribelles report the creation of a single-cell transcriptome atlas for the yeast *Saccharomyces cerevisiae*, called the Yeast single-cell Genotype-to-Transcriptome Atlas (yscGTA-seq). To do this the researchers re-engineered the yeast knockout collection to include transcribed barcodes. They profiled 3,500 mutant strains under normal and osmostress conditions using single-cell RNA sequencing. They find that cells are largely robust to genetic perturbations and primarily respond to environmental change. They identified "intrinsic" cell states (preserved across conditions) and "extrinsic" states (condition-specific). The study identified genetic drivers of transcriptional heterogeneity, finding that most mutations either don't affect heterogeneity or increase it, with few decreasing it. Transcriptional heterogeneity was found to influence cell fitness, with excessive heterogeneity potentially making cells more vulnerable to stress. This work represents a potential advance in understanding the relationship between genotype, transcriptome, and phenotype in yeast. However, there are several issues that the authors should address before the paper is published.

- As presented, the key motivation for this study is to assess the genetic basis of heterogeneity. However, ultimately little is learned about expression heterogeneity and no confirmatory experiments are performed to validate the findings from single cell RNAseq data, whereas additional experiments are performed to assess the effects of stress response expression and aging-associated expression. Given the lack of convincing results regarding the genetic basis of heterogeneity, the authors should consider removing it from the paper and/or strengthening the results with additional experimentation.

- As the authors state, about 25% of non-essential deletions have been characterized using bulk RNA-seq and/or microarrays. The authors should compare the results of their single cell expression analysis with bulk data to confirm that it reproduces known effects of gene deletions.

- It is unclear from the text how good the coverage is per genotype. The authors report 550 genes/cell and 1200 molecules per cell (which is low compared with other studies using scRNAseq in yeast), but it is unclear what the coverage is over the ~ 100 cells/genotype.

- I find it very difficult to interpret the heat map of expression. Does the diagonal line correspond to the expression of the deleted genes? What are the striking vertical lines?

- The authors state that "most mutants displaying little to no differences in differentially expressed genes". This is surprising and I am not sure that it is consistent with prior publications in bulk. Is it possible this reflects low statistical power? Or is this a biologically meaningful result?

- how was the cell cycle regressed and what is the rationale for this? Is there information in the distribution of cell cycle stages for each mutant e.g. with respect to regulators of the cell cycle.

- I am unconvinced by the clustering that is used to define 20 and 18 states. From the UMAP plots there looks like there is minimal structure in the data and it is not clear how robust these clusters are. This is consistent with the authors' finding that many cluster markers are not distinct and a shared across states/clusters.

- what does it mean that a cell expression state is "metastable"? Do the authors mean that it is easily perturbed? This term in this context should be clarified.

- the authors convincingly demonstrate increased expression FIT3 and HUG1 corresponds with increased cell age, as previously shown by Patnaik et al., it is not clear what additional insights come from replication of these previously reported results.

- The authors claim that “mutations that promote phenotypic heterogeneity associate with reduced fitness”, but it is not clear what the basis of this claim is and how one distinguishes this class of mutation from one that simply results in reduced fitness.
- Personally, I find the abbreviation “yscGTA” is not helpful as it is unclear how to pronounce this or if one should simply state each letter.
- Figure 1D indicates huge variation in the number of cells for each genotype. What is the explanation for the extremely abundant genotypes?
- It is unclear what information 1F and 1H convey and they do not depict “correlation number” as stated in the legend.
- The order of genes in 5A and 5B is not explained. Is it the same in both plots?
- Jackson et al., 2020 was the first to report the engineering and use of transcribed barcodes in 3' UTRs for scRNAseq in yeast. The authors have cleverly built on that approach, but I think it should be more clearly stated that the method is based on that original approach.

(Remarks on code availability)
URL does not work.

Version 1:

Reviewer comments:

Reviewer #2

(Remarks to the Author)

We would like to thank the authors for a nice set of new analyses and experiments. We have no further concerns and wish them good luck with the publication.

Three typos:

1. New Extended Data Figure S1D: “at least than 10 cells” -> “at least 10 cells”
2. New Extended Data Figure S1D: “on tope” -> “on top”
3. New Figure S1K: “Per each” -> “For each”

(Remarks on code availability)

Reviewer #3

(Remarks to the Author)

The paper by Nade-Ribelles has been significantly improved through the additional analyses performed by the authors. My comments on the initial manuscript have largely been addressed. The comparison to the Kemmerman et al data provides good validation of the data generated using the scRNAseq approach. It is quite striking how little effect gene deletion has on the transcriptome for the majority of genes (as has been previously observed). The authors might like to discuss/speculate on the interpretation of this. Is it due to gene redundancy or that only direct regulators of transcription have strong impacts. It would be interesting to consider the extent to which cells might adapt (physiologically) to a gene deletion and how these results might differ for an acute perturbation such as a knockdown using CRISPRi.

(Remarks on code availability)

REVIEWER COMMENTS

Reviewer #2 (Remarks to the Author):

The authors engineered a new budding yeast knock-out library where deletion barcodes are expressed as RNA molecules and can therefore be detected by RNA sequencing. This is particularly useful as it opens the way to genetic screens with a read out based on single cell RNA sequencing (scRNA-seq). To showcase this resource the authors performed two experiments during rapid growth and in response to stress (0.4M NaCl, 15 mins). In addition to describing a great resource, this allowed to gain insight into cellular states plasticity and their reliance on genotypes. It also uncovered new knowledge on diverse genes function in single yeast cells as well as on the regulation of phenotypic heterogeneity. It is an interesting paper and I have listed a series of issues below that should be considered before publication.

We would like to thank the reviewer for the constructive and insightful feedback. We have addressed the concerns raised by the reviewer which we believe nicely complement and strengthen the revised version of the manuscript.

Major:

1. Figure 1D: It would be useful to show projections of each axis as density histogrammes as on figure 4E and to report the median in addition to the mean.

As suggested, we have added the marginal histograms and included both the mean and median number of cells in a new Figure 1D. The median number of cells per genotype is approximately 80 in both conditions (77 control, 87 NaCl). We have also specified the mean and median measurements in the main text of the revised manuscript.

New Figure 1D. Correlation of cell number per genotype across conditions. Points are colored by density, warmer colors indicate higher density. Marginal histograms show the distribution of cell numbers for each condition.

2. Figure S1D: I find the label “clone markers” a bit confusing. These are the number of regulated genes compared to control and not number of clones correct? Shouldn't it read “number of regulated genes”? Also, is it really accurate to say that most mutants are in the first bin? Considering the tail the number of clones in the (0,1) bin and those in all other bins seems quite balanced. This need to be clarified.

We agree with the reviewer that the term “clone markers” could lead to confusion; we have relabeled this with “number of regulated genes” as suggested.

Regarding the distribution of mutants, the reviewer observation is correct as in about half of the genotypes there are no differentially expressed genes between clones; number of DE = 0; 174 genotypes versus FC>0; in 194 genotypes. Our initial histogram was meant to provide a detailed distribution of the number of regulated genes by using bins of 1 gene which overestimate the differences. In fact, only 23% of clones have more than 3 differentially expressed (DE) genes, suggesting that the differences between clones are small which was our initial message. We have now included the data taking into account the new threshold of 3 differentially expressed genes and we have clarified this point in the main text and methods sections as suggested by the reviewer.

New Extended Data Figure S1D. . Histogram the classification of clones based on the number of differentially expressed (FC>2 pval <0.05). Only genotypes with at least than 10 cells per clone. Pie chart on tope, depicts the percentage of clones that express >3 differentially expressed genes (n=365). Bars and pie chart are colored according to the number of differential expressed genes and dashed line indicates the threshold.

3. Figure 1J-L: The authors state that the stress response genes are driving the split of the cells from the two conditions on the UMAP into distinct populations. How about a technical batch effect? The design with 29 libraries seems robust to this but it might be good to comment.

We acknowledge the reviewer’s concern on potential batch effects on such a huge experimental set up. We would like to note that we carefully designed a balanced experimental workflow to minimize batch effects. Briefly, cells were harvested, immediately fixed with methanol, and frozen in aliquots. To generate the libraries we used the Singleron Matrix equipment that automatically loads cells and performs an on chip cell lysis and mRNA capture. The equipment has 2 slots, for each batch we ran control and stress sample in each slot, alternating the sample positions between slots for each run. Libraries were processed up to cDNA amplification, then stored for joint processing. Remaining library preparation steps were performed simultaneously for all samples. All 29 runs were sequenced as a single pool in a NovaSeq S4 lane to further reduce potential batch effects during sequencing.

To assess the batch effect, we clustered the raw data from the different batches using a PCA analysis. As expected, we found a clear condition-dependent clustering. We have included this information in the revised manuscript and improved the description of the experiment in the methods section.

New Figure S1K. Principal component analysis of the entire dataset. Per each batch and condition the average value for PC1 and PC2 are represented.

4. Figure 1L: “we could detect similar cell states within each condition...”. Does this figure really show that states are matching?

We thank the reviewer for pointing this out. We apologize for this error, we agree that Figure 1L does show that the states are matching. We meant to say that cell states were dominant over genotypes. The equivalence of cell states is addressed in Figure 3. We have modified the text to reflect the intended message.

5. Are any of the states enriched for specific cell-cycle stages? This could be extracted from the scRNA-seq data and would help connecting this work with a large body of yeast literature.

This is an interesting issue raised by the reviewer. To prevent clustering effects due to cell cycle genes, we regressed cell cycle from the state analyses. This is a standard procedure of scRNA-seq analysis in order only assess the effect at the transactional level of genetic or environmental perturbations other than the expected cell cycle effects. Actually, to visualize the relevance of cell cycle regression, we projected the cell cycle scores onto the UMAP before and after cell cycle regression. As expected, after cell cycle regression cells distribute more uniformly across the UMAP space.

However, we fully agree with both reviewers that the data on cell cycle is potentially interesting for many researchers and thus, we have now included the cell cycle distribution per genotype (new Extended Data Table 4; and new Extended Data Figure S2E and S2F). We have also updated the manuscript, code, objects and database to enable researchers to explore the dataset before and after cell cycle regression.

New Extended Data Figure S2E and S2F. E-F) Barplot represents the percentage of cells in each cell cycle phase for each cell state cluster for control (E) or NaCl (F) dataset .

6. Figure 2E-F: I believe the control population here is problematic. Top 2% expressing cells for aging markers are compared to randomly sorted cells. Levels of gene expression correlate globally with features like cell size or metabolic status. To take this into account the control should be the 2% higher expressors of the constitutive reporter and not a random set.

This is a very nice suggestion. Our initial experiment intended to show that these cells represent the oldest within the population and this is why we added a random sort of the population. However, we agree that adding the suggested control would strengthen the data.

As suggested, we have included a control with the top 2% of high expressing cells using a new reporter (*SAG1*), which is a cluster-specific marker unrelated to any of the aging subtypes. We sorted the top 2% of cells *SAG1* expressing cells and include *HUG1* as a control and, assessed their aging by measuring both bud scar count and mitochondrial morphology patterns. The top 2% *SAG1* expressing cells did not display any significant signs of aging in the number of scars or mitochondrial morphology resembling wild type cells, whereas the top 2% *HUG1* and *FIT3* expressing cells consistently displayed higher number of scars and diminished tubular mitochondrial morphology. The absence of aging indicators in the *SAG1* cells further supports the specificity of this aging phenotype and strengthens the cell state-phenotype relationship. We have now modified Figures 2E and 2F to include the new data and modified the text accordingly.

New Figures 2E-2F. E) Number of scars per 100 cells determined by Calcofluor White Stain for the top 2% expressing cells of the indicated reporters. Data represents the mean of three independent replicates. Paired t.test is shown against the wild type (n=3). F) Distribution of mitochondrial morphologies for the indicated strains under control conditions by MitoTracker staining (n=3).

7. Figure 3: The use of “intrinsic” and “extrinsic” is a bit confusing. Some states could result from molecules secreted by cells within the culture. Such states would result from responses to external molecules as with the NaCl treatment but in this case would be called intrinsic. Do I make sense? This needs clarification.

We had not realized that the use of “intrinsic/extrinsic” could be misleading. The message we were trying to convey was that we found both states shared across conditions (originally named “intrinsic”) as those cell states that were defined by the similar genes and “extrinsic” as those that did not share similar cell state markers. However, following the reviewer’s suggestion and to make the concept clearer to the readers, we followed the reviewer’s advice and renamed the cluster as “core” for those constant across situations and “responsive” for those modulated by adaptive responses.

8. Figure 3B: “For example, under stress, markers of the hypoxic aging and phosphate metabolism clusters (4C-15C and 9C, respectively) were widely expressed...”. This is possibly due to image quality, but to me it seems that those markers disappear upon stress induction instead. This needs clarification.

In our initial sentence by “widely expressed”, we meant to say that the expression becomes delocalized/homogeneous and therefore its expression no longer clustered. However, this is a very valid point raised by the reviewer. We have now assessed the expression of cluster 9C (phosphate metabolism), 4C and 15C (hypoxia and late-hypoxic aging) across all clusters in each condition and showed that their mean expression is slightly lower upon stress (see below). Therefore, we have improved the phrasing to make this point clearer.

Rebuttal Figure 1. A-B) Violin plots represent the mean expression of the phosphate cluster (9C) as a function of cell state in control (A) or NaCl (B) condition. For each cluster the mean expression is shown as a black horizontal line and the population mean is shown as a dashed line. C-D) Expression distribution of the hypoxia aging markers from cluster 4C represented as in A-B. E-F) Expression distribution of the hypoxia #2 signature of the late-stage hypoxia aging (cluster 15C) for the indicated clusters and conditions as in A-B.

9. Figure 3C: On the figure, states have most of the time more than one counterpart in the other condition. Doesn't it challenge the statement that states are shared between conditions? To me it suggests that states are mostly different between control and stress with partial overlap. This needs clarification.

The message we were trying to convey was that we found both states shared across conditions (originally named "intrinsic") or responsive by extrinsic factors like stress (originally named "extrinsic"). We have followed the reviewer's suggestion and improved our cluster classification to rename the clusters as "core clusters" for those that are reciprocally detected in both conditions and "responsive clusters" those that change upon stress. Our initial analysis assessed the cluster equivalence based only on the expression of upregulated cell state markers. However, we agree with the reviewer that for some clusters we observed one or more counterpart in the other condition. This sometimes occurs because interrelated clusters in control conditions (e.g., pheromone clusters 2C, 12C, and 16C) are mainly represented by fewer clusters in NaCl (1N and 15N), for instance causing both to correlate with cluster 12C. One explanation is that our initial analysis uses upregulated cluster marker genes to assess correlation.

To improve this point, we employed a quantitative approach using the label transfer function built into the Seurat package. This unbiased method identifies cell pairs in the same biological state across different conditions, comparing the anchor reference sample (control) to the query (NaCl) dataset. This analysis revealed that 16 out of the 20 control states share at least 40% of their cells with a corresponding stress state. These results are very similar to our initial classification using only the expression of upregulated cell state markers where we identified 15 out of 20 states as shared across conditions (Figure 3C). This method offers a clearer picture of cluster relationships across conditions, we believe this analysis agrees with our initial interpretation and provides solid evidence for the existence and both stable "core states" but also dynamic "responsive clusters". We have renamed cell states in the main text, included the label transfer as a new Extended Data Figure S3A and the complementary table with the frequency of cells in each predicted cell state for both control and stress condition Extended Data Table 5.

New Figure S3A. A) The alluvial plot represents percentage of corresponding cells pairs across clusters and condition obtained from performing a label transfer. Clusters identified in the control condition (left) were used as a reference to predict cell type labels in the stress (right). The streams are colored based on the control clusters and indicate the connectivity across the indicated clusters.

10. Figure 3F: Same comments as in point 5 applies here.

We understand that the reviewer refers to point 6 where he/she suggested to add another control for the top 2% of the population. As suggested by the reviewer, we sorted the top 2% population of wild type cells constitutively expressing *SAG1*, the same reporter gene used before (point 6) as a control. Reassuringly, the top 2% *SAG1* expressing cells did not alter competitive fitness when compared to a wild type strain (grey). This reinforces the idea that the transient expression of cell state specific phenotype genes such as *HSP12* leads to a stress-resistance phenotype. We have now included this control in the revised version of the manuscript (new Figure 3F).

New Extended Data Figure S3C. Cells with the top 2% expression of the SAG1 mCherry reporter (pSAG1-UbiM-mCherry-tSAG1) were sorted and grown in combination with wild type cells (labelled with GFP) in rich media (control) or in the presence of stress (NaCl). Growth of the two populations was assessed at time 0 or after 48 hours by flow cytometry. Data represents mean and standard deviation of three independent experiments. Paired t-test comparing time 0h and 48h of each condition is shown on top.

11. P9: "...is independent of cell number (Extended Data Fig. S5B. S5C).". I am not sure I see this on the figure. To me it looks like the scores ≥ 1.4 (used as cut off on 5A and B) are clustered on the left of the graph in strains with lower number of cells (< 64 cells). Am I missing something? If not, more validation is required to ensure that higher scores observed in certain mutants are not a result cell number and that biological conclusions are supported by the data.

We appreciate the reviewer's observation regarding the clustering of higher scores in strains with lower cell numbers. Indeed, this pattern is consistent with the typical behavior observed in MA plots, where lower cell/counts often display increased variability. Our analysis follows the methodology employed in the human genome-scale perturbation screen by Replogle et al., 2022. In that study, which had a comparable overall coverage (mean 100 cells/genotype), they did not apply corrections to the leverage scores depending on cell number. However, to bypass the limitation of the leverage score in strains with very few cells, we filtered out genotypes with less than 6 cells. Of note, we included seven wild-type clones in our study with a different range of cell numbers from low (10 cells), average (87 cells) to high (175 cells), and they do not show differences in the leverage score. Therefore, albeit the reviewer observation is correct in the sense that some of the high heterogeneity mutants have fewer cells than average, we ensured comparability of results across studies and added an additional filter to reduce this limitation. Following the reviewers' suggestion, we have now clarified this aspect in the revised manuscript.

Minor:

1. Figure S1B: Should one of the D2 labels be D1?

We thank the reviewer for pointing this out. Indeed, one of the labels should be D1, we have fixed it in Figure S1B.

2. P7: "Our results hence indicate that phenotypes can be promoted by alterations in gene function and highlight the potential of using transcriptional phenotypes to define novel phenotype-genotype interactions.". As it is, this statement reads a bit trivial? Maybe rephrase?

As suggested by the reviewer we have rephrased this point to: "Our results demonstrate that gene deletions can lead to the accumulation of cells in specific cell states, characterized by distinct expression

patterns. Although most cell states are shared across genotypes, mutations influence their relative abundance. This information allows us to leverage the similarity in transcriptional phenotypes to link functionally related genes and elucidate phenotype-genotype interactions.”

3. P.8: “we have defined the regulatory logic that governs cell state”. This is a very strong statement. What do the authors mean here exactly?

We meant to say that our dataset reveals a set of genes associated to cell states, some of which we experimentally validated. Additionally, we wanted to highlight that by using the cell state distribution, we identified underlying mutations that promote cell state attraction or deplete cells from a state. However, we agree that our original statement was overly simplified and we have now modified the text to ease the understanding of our initial meaning.

Reviewer #3 (Remarks to the Author):

In this paper Nadal-Ribelles report the creation of a single-cell transcriptome atlas for the yeast *Saccharomyces cerevisiae*, called the Yeast single-cell Genotype-to-Transcriptome Atlas (yscGTA-seq). To do this the researchers re-engineered the yeast knockout collection to include transcribed barcodes. They profiled 3,500 mutant strains under normal and osmotic stress conditions using single-cell RNA sequencing. They find that cells are largely robust to genetic perturbations and primarily respond to environmental change. They identified "intrinsic" cell states (preserved across conditions) and "extrinsic" states (condition-specific). The study identified genetic drivers of transcriptional heterogeneity, finding that most mutations either don't affect heterogeneity or increase it, with few decreasing it. Transcriptional heterogeneity was found to influence cell fitness, with excessive heterogeneity potentially making cells more vulnerable to stress. This work represents a potential advance in understanding the relationship between genotype, transcriptome, and phenotype in yeast. However, there are several issues that the authors should address before the paper is published.

We thank the reviewer for his/her constructive feedback on our manuscript and appreciate his/her suggestions that have served to improve the manuscript.

1- As presented, the key motivation for this study is to assess the genetic basis of heterogeneity. However, ultimately little is learned about expression heterogeneity and no confirmatory experiments are performed to validate the findings from single cell RNAseq data, whereas additional experiments are performed to assess the effects of stress response expression and aging-associated expression. Given the lack of convincing results regarding the genetic basis of heterogeneity, the authors should consider removing it from the paper and/or strengthening the results with additional experimentation.

We had several motivations to perform this study. First, we aimed to design and construct a system that would enable to perform isogenic genome-scale deletion experiments to assess scRNA-seq. Second, we wanted to generate a global atlas of yeast genotype-transcriptome-phenotype that would represent the largest and high resolution (single cell) transcriptome atlas. To showcase the power of our dataset we used to catalogue, for the first time, the repertoire of cell states and to identify the effect of genetic perturbations. Thus, we believe this demonstrates that our dataset can be integrated with the large amount of information available for the Yeast Knock Out Collection. To our knowledge this is the first, genome-scale scRNA-seq performed simultaneously under control and stress conditions.

Yeast is an ideal organism for studying the regulators of transcriptional heterogeneity due to its unique characteristics. Unlike higher eukaryotes, yeast cells are unicellular organisms that have a unified cell identity and are isogenic in nature. Additionally, yeast exhibits highly homogeneous signaling dynamics (Pelet et al., 2011 Science) and lacks cell-to-cell communication in haploid cells of the same mating type, providing a clean molecular context to isolate and analyze transcriptional variations at an organism scale. Hence, we harnessed our dataset to uncover regulators of transcriptional heterogeneity under control and stress conditions. Our study has revealed that drivers of transcriptional heterogeneity are functionally diverse and condition-dependent. In addition, we used previously published phenotypical data on those mutant cells with higher/lower heterogeneity to relate transcriptional heterogeneity to phenotype. Therefore we felt that using already validated experimental data provides an experimental validation to our study. In addition, the paper from Repogle et al., 2022 in *Cell* displayed a similar type of analysis for transcriptional heterogeneity and found some common activities such as the basal activation of the stress response and severe transcriptional phenotypes through mitochondria dysfunction like those identified

in our study. Of note, our study is more comprehensive and performed in an isogenic strain with full deletions instead of a drop out analysis. While we agree that our dataset does not provide defined molecular mechanisms promoting heterogeneity, we have been able to identify a number of functions and genes (in two different conditions simultaneously) that when altered have an impact on transcriptional heterogeneity, which to our knowledge has not been assessed before in any organism at this scale and resolution. Thus, we believe that the identification and classification of transcriptional represents a novel observation and resource that is of interest to the community to maintain it in the manuscript.

2- As the authors state, about 25% of non-essential deletions have been characterized using bulk RNA-seq and/or microarrays. The authors should compare the results of their single cell expression analysis with bulk data to confirm that it reproduces known effects of gene deletions.

As suggested by the reviewer, we compared our single cell RNA-seq data to the non-essential deletions performed by (Kemmeren et al., 2014 Mol. Cell). Because the original study performed transcriptomic profiling under basal conditions, we performed the comparison between genotypes profiled in our control condition. To this end, first we examined the correlation between the number of differentially expressed genes per genotype in the Kemmeren et al. microarray study and our scRNA-seq dataset. Despite substantial technical, experimental, and analytical differences, we observed a consistent pattern in differential expression across both studies (new Extended Data Figure S1I), which is proportional to the mutant signal/responsiveness in each approach. Furthermore, to evaluate global transcriptional similarity, we correlated the fold changes between studies. Again, we found a positive correlation in gene expression changes, indicating that the transcriptome differences are effectively captured by the scRNA-seq dataset (new Extended Data Figure S1J).

Although a comparable dataset for osmostress is not available, we ranked the expression of the mean expression of induced Environmental Stress Response (iESR) genes to evaluate the behavior of mutants expected to show impaired induction based on existing literature.. We plotted the positions of key transcription factors involved in iESR, such as MSN2, alongside osmostress-specific factors like Sko1 and Hot1. Additionally, we examined mutants from the High Osmolarity Glycerol (HOG) pathway, specifically *hog1* and *pbs2* mutants, which encode the effector MAPK and MAP2K, respectively. Notably, all these mutants exhibited behaviors consistent with the expected diminished induction of the iESR reported in previous bulk transcriptomic analyses.. These findings show that our dataset accurately captures known transcription patterns, which agree with previous datasets in control and osmostress and reinforces the potential to assess unexplored transcriptional phenotypes. We have added this information to in the revised manuscript (Extended Data Figure S1I and S1J) and modified the text accordingly.

New Extended Data Figure S11 and S1J. I) Scatter plot showing the median (log) number of differentially expressed genes in control conditions for the intersecting genotypes with the large-scale deletion microarray profiling from Kemmeren et al., 2014. Each point shows a genotype. J) Expression ranking of the mean iESR signature (y axis) across genotypes (x axis) under stress condition. Genotypes are represented in descending order and reference genotypes (WT) and mutants reported in the literature to display impaired induction of the response are highlighted and labeled in red.

3- It is unclear from the text how good the coverage is per genotype. The authors report 550 genes/cell and 1200 molecules per cell (which is low compared with other studies using scRNAseq in yeast), but it is unclear what the coverage is over the ~100 cells/genotype.

Our dataset indeed has an average of 550 genes/cell and 1200 molecules/cell. The gene coverage on scRNA-seq is strongly dependent on the library type and there is an inevitable trade off with the number of cells. While it is true that these numbers are low when compared to low throughput methods (Nadal-Ribelles et al., 2019 and Gasch et al., 2017) these studies profiled less than 200 cells each and therefore could reach a much deeper sequencing depth (1M reads/cell in Nadal-Ribelles et al 2019), which is not realistic nor compatible with HT analyses such as our dataset. Moreover, the use of droplet based, combinatorial (split-seq) or microwell based strategies are known to consistently generate libraries with fewer genes and molecules. We would like to highlight that the coverage of 550 genes is close to the recently published studies including the manuscript from Jackson et al. 2020, in which the authors profiled 12 transcription factor deletions across conditions (684 gene for 38K cells using diploid cells which are known to have higher number of molecules than haploids). Similarly, other large-scale reports in yeast such as Dohn et al., 2022 reported 35K cells and a range of 193-967 genes/cell and Brettner et al., 2024 reported 43K cells and 100-600 genes (reviewed in Nadal-Ribelles et al., 2024). . Therefore, we believe the coverage and resolution of our dataset falls within the standard resolution of experiments of this size and complexity.

However, following the reviewer's suggestions, we have now included the mean and median cell numbers for each genotype and condition (**see referee #2 question 1**) that confirms that each genotype is represented by ~100 cells/genotype. We have also clarified this part of the text in the revised manuscript.

4-I find it very difficult to interpret the heat map of expression. Does the diagonal line correspond to the expression of the deleted genes? What are the striking vertical lines?

The reviewer is correct in the interpretation of the heatmap. Yes, the diagonal represents the expression of the deleted gene. The vertical lines represent genotypes with overall higher expression. We have modified the figure legend to make this point clearer.

5- The authors state that “most mutants displaying little to no differences in differentially expressed genes”. This is surprising and I am not sure that it is consistent with prior publications in bulk. Is it possible this reflects low statistical power? Or is this a biologically meaningful result?

The meaning of our statement was that most of the mutants in the collection did not display great changes in transcriptional profiles when compared to wild type. It is true, however, that this statement underestimates the effect of certain mutations that alter the expression of a few genes.

The bulk microarray dataset of 25% of the YKOC classified mutants as “responsive” if 4 differentially expressed genes were detected using an expression threshold of $FC > 1.7$ and $pval < 0.05$. Using this criteria, and keeping in mind that the mutant selection was biased towards transcription-related mutants, the authors reported 53% of the assayed mutants as responsive. Importantly, if we use the same threshold in the number of differentially expressed genes to of 4 nearly 75% of mutants are also responsive in our dataset.

It is worth to notice that our classification reflects a more stringent criteria rather than the lack of statistical power. The main two differences in criteria are: first, we have followed the standard fold change criteria of $FC > 2$ and $pval < 0.05$, which sets a more precise threshold and second, we defined a higher threshold of ≥ 10 differential expressed genes to consider a mutant responsive instead of 4 genes used by Kremmeren et al., 2024 (original Figure 1D and 1F). Our reasoning is that a higher threshold of (≥ 10 genes) would better reflect responsive mutants than only using 4 genes. Therefore, the apparent discrepancy from our initial statement is due to differences in thresholds and classification criteria, rather than an issue with the data. However, to make the data clearer to the readers we generated detailed binning criteria to group genotypes every 5 DE genes.

We have now discussed this in the revised manuscript and as suggested in Reviewer #3 question 2 included the comparison with bulk studies (new Extended Data Figure S1I, S1J).

6- how was the cell cycle regressed and what is the rationale for this? Is there information in the distribution of cell cycle stages for each mutant e.g. with respect to regulators of the cell cycle.

This is a very valid point, also raised by reviewer #2. To perform the cell cycle regression, we used the default Seurat pipeline (see Reviewer 2-Question #5). Briefly, to assign the cell cycle phases we used the “*CellCycleScoring*” function from Seurat to classify cells in the corresponding cell cycle phase using expression of canonical phase specific genes. Then we regressed out the cell cycle scores by supplying the cell cycle variable genes to the “*vars.to.regress*” argument. using the calculated “S.Score”, “G2M.Score”.

Cell cycle is a widely known factor that introduces significant variability in gene expression patterns that can prevent to identify biological differences and it is a standard procedure across scRNA-seq workflows. The main aim of our study was to generate a perturbation-based single-cell Genotype-to-Transcriptome Atlas. Because our experimental design was intended to only profile the effect of the genetic and environmental perturbations we eliminated the effect caused by cell cycle genes. However, we fully agree with both reviewers that including the information on cell cycle can be very useful for the community and as such, we have now included enabled to explore the data with or without cell cycle normalizations.

Furthermore, we have now added and Extended Data Table 4, which provides the number and percentage of cells in each cell cycle phase per genotype for both conditions as suggested by the reviewer.

Extended Data Figure S1L. UMAP of the entire dataset before (upper panels) or after cell cycle regression (lower panels) using the CellCycleScoring function from Seurat. Cells are colored based on the assigned cell cycle phase (shown on top).

7- I am unconvinced by the clustering that is used to define 20 and 18 states. From the UMAP plots there looks like there is minimal structure in the data and it is not clear how robust these clusters are. This is consistent with the authors' finding that many cluster markers are not distinct and a shared across states/clusters.

UMAPs are useful for data visualization and Louvain clusters were calculated using standard Seurat pipeline using a resolution= 1. It is true that cluster resolution can be modified to define a different number of clusters within one sample, however we have used a standard normalization (SCT transform) and clustering algorithm (Louvain clustering) both ubiquitously used across scRNA-seq datasets. The apparent lack of structure in the UMAP plot mentioned by the reviewer, does not negate the existence of biologically meaningful clusters.

Moreover, our results strongly agree with those reported by Jackson et al. 2020, in which they analyzed the deletion of 12 transcription factors and found that they did not display a structure other than condition-specific clustering. We believe however there is a biological meaning to this structure which is related to the intrinsic properties of *S. cerevisiae*. Because yeast cells have a unified cell type / identity we expected the UMAP to display lower complexity than those with different cell types. Our data demonstrates that clusters represent a continuum of cell states in which for example daughter cells will transition to mother cells and eventually these will reach the aged population. Even though, we find shared markers across clusters, the co-expression of two genes specifically labels each cluster which suggests the cluster signature are indeed unique. Of note, we want to emphasize that taking together the

clusters that reflect cell states known in the literature, we also experimentally validated some of these states through functional assays, which adds robustness to the study.

8-what does it mean that a cell expression state is “metastable”? Do the authors mean that it is easily perturbed? This term in this context should be clarified.

By metastable we meant to say that it is shared across most genotypes and therefore we believe represents a core state. We have clarified the text the use of the term.

9- the authors convincingly demonstrate increased expression *FIT3* and *HUG1* corresponds with increased cell age, as previously shown by Patnaik et al., It is not clear what additional insights come from replication of these previously reported results.

The data from Patnaik et al. 2022, used a laboratory aged yeast cell culture to perform bulk RNA-seq experiments. Through this approach they identified a transcriptional signature specific to old cells. Remarkably, this transcriptional signature is the sum of the two patterns identified in the scRNA data that includes independently the expression of *FIT3* and *HUG1*. This is the beauty of the data generated, the single cell resolution of our dataset has allowed us to resolve the bulk aging signature into two distinct transcriptional programs underlying two different aging subtypes via modular expression of iron (represented by *FIT3*) or hypoxic genes (*HUG1*). The replication of their results with the addition of both programs demonstrates the robustness of our dataset and it clearly shows the advantage of having the scRNA resolution. In addition, by looking at cell state distribution across the genotypes of the different mutants in the collection, we were able to reveal new regulators of iron metabolism other than the expected mutants with impaired mitochondrial function, such as mutants in the ESCRT complex. Finally, the stress dataset allowed us to demonstrate that iron-aging is stable across conditions while hypoxic signature is lost indicating that different conditions can lead to the differential regulation of gene expression programs that lead to specific phenotypes. Therefore, our study provides a deconvolution of the bulk signature defined in bulk while expending the genes signature and linking gene function to phenotype. We have emphasized this aspect on the revised manuscript.

10- The authors claim that “mutations that promote phenotypic heterogeneity associate with reduced fitness”, but it is not clear what the basis of this claim is and how one distinguishes this class of mutation from one that simply results in reduced fitness.

We believe that the reviewer confused the term phenotypic heterogeneity for phenotypic *homogeneity* in the original manuscript. We meant to refer mutants that are enriched in specific cell states (phenotypic homogeneity). Our original statement was: “mutations that promote phenotypic *homogeneity* associate with reduced fitness.”. This is based on the observation that mutants showing enrichment towards certain cell states (cell state attractors) exhibit reduced fitness under various conditions, as reported in the 14 stressor panel in Constanzo et al., 2021 Science (Figure 3J and 3K). However, based on the comment by the reviewer, we realized that the term “phenotypic homogeneity” might be confusing and therefore we have rephrased the statement to enhance clarity.

11- Personally, I find the abbreviation “yscGTA” is not helpful as it is unclear how to pronounce this or if one should simply state each letter.

As suggested by the reviewer, we have removed the abbreviation.

12- Figure 1D indicates huge variation in the number of cells for each genotype. What is the explanation for the extremely abundant genotypes?

Fluctuations in the number of cell per mutants is a common event in perturbation experiments. Not all mutants grow at the same rate or might even lyse at the same efficiency and therefore it is expected the number of cells per mutant is variable. To limit those effects, all mutants were grown individually to avoid competition between strains while growing, while we tried to control for an even number of cells at exponential phase, this is technically impossible at an experiment of such scale and given that mutants have different growth rates. This is why we removed genotypes with less than 6 cells. Of note, one of the most abundant genotypes in Figure 1D represents the wild type strain all together (n=500 cells), which we consciously spiked at a higher number to have a good reference for the whole study.

13- It is unclear what information 1F and 1H convey and they do not depict “correlation number” as stated in the legend.

We meant to use this graph to first provide an overall view of the expression trend generated by the profiled genetic perturbations, which as mentioned above in control conditions tend to lead to upregulation of genes (recapitulating Kremmeren et al., 2014). We thank the reviewer for noticing the mistake. Panels 1F and 1H represent the comparison of induced and repressed genes for each genotype in each condition. We have corrected the figure legend accordingly.

14- The order of genes in 5A and 5B is not explained. Is it the same in both plots?

We appreciate the observation and acknowledge that our initial explanation was insufficient. Regarding the x-axis order, in panels 5A and 5B, is indeed the same. The genotypes are plotted in alphabetical order in the x axis (from left to right) with the wild type strain plotted last. We have clarified this aspect in the figure legend.

15- Jackson et al., 2020 was the first to report the engineering and use of transcribed barcodes in 3' UTRs for scRNAseq in yeast. The authors have cleverly built on that approach, but I think it should be more clearly stated that the method is based on that original approach.

We acknowledge that our strategy is inspired by the use of transcribed barcodes at the 3'UTR, and cited the work by Jackson et al. in the initial version of the manuscript. However, we agree that we could have stated it more clearly. We have now clarified this aspect in the revised manuscript. Of note, we would like to point out that our strategy significantly differs from the one reported in Jackson et al., by several reasons. The initial report by Jackson et al., required to generate a specific deletion cassette for each mutant and its replicates. While, undoubtedly powerful, scaling this design to a large number of mutants would be expensive and extremely labor intensive. Our approach relies on a simpler design that uses a single PCR cassette to convert the existing YKOC barcodes for scRNA-seq compatibility in a one-step transformation. The resulting library also includes a novel clone barcode that served multiple purposes: it enables tracking and performance of replicates while allowing clonal tracing in scRNA-seq within the same experiment. Additionally, it extends the functionality of the original collection by enabling clonal phenotyping in bulk using common primers upstream of the URA3 STOP codon and D1 primer. As such, besides the reduced complexity in generating the library, this new library and dataset preserve the structure of the YKOC allows to seamlessly integrate with the vast amount of data already generated over decades by the yeast community.

We would like to mention that from an experimental and methodological point both studies are different. We used a microwell-based scRNA-seq, which had not been used in yeast to make the library cost more efficient than commercial droplet-based methods. We also incorporated a methanol-fixation method which we have used before in low throughput methods (Nadal-Ribelles et al., 2019) to preserve the fast-responding yeast transcriptome. This allows to load intact cells to perform an on chip digestion rather than using spheroplasts as an input as reported by Jackson et al. 2020. Following the reviewer recommendation, we have now modified the text to clearly state the contribution by Jackson et al., 2020 and discussed the key differences between approaches.

16-Reviewer #3 (Remarks on code availability) URL does not work.

We apologize for the URL not working, we have now reviewed the Array Express, and uploaded the code and processed data to Zenodo. If the manuscript is accepted, we will upload the processed data and code to Dryad (Nature recommended data repository) for a more friendly use of the databases.

Please find the update reviewer tokens below (both datasets are uploaded as private/restricted access and accessible with the provided tokens below).

Array Express:

Raw sequencing data of the scRNA Atlas;

token:

<https://www.ebi.ac.uk/biostudies/arrayexpress/studies/E-MTAB-14004?key=152436a0-d3e6-484e-9653-0779be2e00db>

Zenodoo: 10.5281/zenodo.14062629

Code and processed Seurat objects (split by condition or joint dataset).

<https://zenodo.org/uploads/14062629>;

token:

<https://zenodo.org/records/14062629?preview=1&token=eyJhbGciOiJIUzUxMiIsImIhdCI6MTczMTI0OTkzOCwiZXhwIjoxNzYzNTk2Nzk5fQ.eyJpZCI6IjU0ZjY2N2Y2LTM3NjYtNDI0My1hNGY1LWVjZDU0MTc2ZTFmZSIsImRhdGEiOnt9LCJyYW5kb20iOiI5NzVINmJiZGRiNTVhYjRhMDYzNTFkN2lwNDZmZGM2MyJ9.-DagtD5YsI5Wi0QmSdX55jnIheSlxqiF-EpAeGvpppP4iW0zcSTLPI0NtnyG2Rk2x28QhJ IoHTOafKHvNcig>

REVIEWERS' COMMENTS

We would like to thank the reviewers for carefully reading our manuscript and their positive recommendation for publication. The feedback provided by both reviewers has led us to include additional analyses and discussions, which have significantly enhanced the quality of the manuscript.

Reviewer #2 (Remarks to the Author):

We would like to thank the authors for a nice set of new analyses and experiments. We have no further concerns and wish them good luck with the publication.

Three typos:

1. New Extended Data Figure S1D: “at least than 10 cells” -> “at least 10 cells”
2. New Extended Data Figure S1D: “on tope” -> “on top”
3. New Figure S1K: “Per each” -> “For each”

Thank you for spotting the typos, we have now corrected the three typos and reviewed for additional mistakes.

Reviewer #3 (Remarks to the Author):

The paper by Nade-Ribelles has been significantly improved through the additional analyses performed by the authors. My comments on the initial manuscript have largely been addressed. The comparison to the Kemmerman et al data provides good validation of the data generated using the scRNAseq approach. It is quite striking how little effect gene deletion has on the transcriptome for the majority of genes (as has been previously observed). The authors might like to discuss/speculate on the interpretation of this. Is it due to gene redundancy or that only direct regulators of transcription have strong impacts. It would be interesting to consider the extent to which cells might adapt (physiologically) to a gene deletion and how these results might differ for an acute perturbation such as a knockdown using CRISPRi.

This is a very interesting insight from the reviewer. Indeed, both the bulk and our new single cell atlas support the idea that only a few deletions have strong effects on the transcriptome. We agree with the reviewer that this needs to be discussed. Although we do not know the cause, we have included in the discussion the potential role of gene redundancies. We have also highlighted the differences between gene deletions and CRISPR knockdowns. The type of genetic modification is one of the main differences compared to previous genome-scale Perturb-seq studies. Gene deletions generate a complete, permanent, isogenic modification, but on the other hand, it might enable for the appearance of compensatory mechanisms. We have now included these points in the discussion.